# Disentanglement via Latent Quantization

**Kyle Hsu**[†] **Will Dorrell**[‡] **James C. R. Whittington**[†§] **Jiajun Wu**[†] **Chelsea Finn**[†]
[†]Stanford University    [‡]University College London    [§]Oxford University
{kylehsu,jiajunwu,cbfinn}@cs.stanford.edu    {dorrellwec,jcrwhittington}@gmail.com

## Abstract

In disentangled representation learning, a model is asked to tease apart a dataset's underlying sources of variation and represent them independently of one another. Since the model is provided with no ground truth information about these sources, inductive biases take a paramount role in enabling disentanglement. In this work, we construct an inductive bias towards encoding to and decoding from an organized latent space. Concretely, we do this by (i) quantizing the latent space into discrete code vectors with a separate learnable scalar codebook per dimension and (ii) applying strong model regularization via an unusually high weight decay. Intuitively, the latent space design forces the encoder to combinatorially construct codes from a small number of distinct scalar values, which in turn enables the decoder to assign a consistent meaning to each value. Regularization then serves to drive the model towards this parsimonious strategy. We demonstrate the broad applicability of this approach by adding it to both basic data-reconstructing (vanilla autoencoder) and latent-reconstructing (InfoGAN) generative models. For reliable evaluation, we also propose InfoMEC, a new set of metrics for disentanglement that is cohesively grounded in information theory and fixes well-established shortcomings in previous metrics. Together with regularization, latent quantization dramatically improves the modularity and explicitness of learned representations on a representative suite of benchmark datasets. In particular, our quantized-latent autoencoder (QLAE) consistently outperforms strong methods from prior work in these key disentanglement properties without compromising data reconstruction.

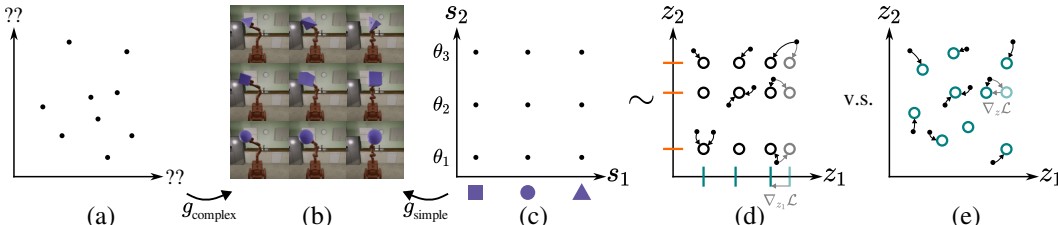

Figure 1: Motivating observations and illustration of the inductive bias of latent quantization. The mapping between true sources (c) and data (b) is simpler than most other possible generative functions from unstructured spaces (a). To help recover the sources, we use quantized latent codes (d)—continuous codes (dots) are mapped (via arrows) to their nearest discrete codes (black circles), each of which is constructed combinatorially from per-dimension scalar codebooks (turquoise and orange ticks). One way this design's inductive bias manifests is in how the codes change from previous values (grayed) as a result of codebook optimization: the combinatorially defined codes move in lockstep, changing the quantized representations of many datapoints. In contrast, naively quantizing into individual vector embeddings (e) would result in codebook optimization having a comparatively local effect.

37th Conference on Neural Information Processing Systems (NeurIPS 2023).

# 1 Introduction

Our increasing reliance on black-box methods for processing high-dimensional data underscores the importance of developing techniques for learning human-interpretable representations. To name but a few possible benefits, such representations could foster more informed human decision-making [62, 27], facilitate efficient model debugging and improvement [16, 44], and streamline auditing and regulation [51, 14]. In this context, disentangled representation learning serves as a worthwhile scaffolding: loosely speaking, its goal is for a model to tease apart a dataset's underlying sources of variation and represent them independently of one another.

Accomplishing this, however, has proven difficult. Conceptually, the field lacks a formal problem statement that resolves fundamental ambiguities [30, 47] without overly restrictive assumptions (reviewed in Section 6); methodologically, evaluation metrics have been found to be sensitive to hyperparameters, ad hoc, and/or sample inefficient [10]; and empirically, there remains a need for an inductive bias that enables consistently good performance in the purely unsupervised setting.

In this work, we answer the call for a better inductive bias for disentanglement. Our solution is motivated by observing that many datasets of interest are generated from their sources in a compositional manner, which entails a neatly organized source space (Figure 1). This distinguishing property of realistic generative processes applies to real world physics as well as human approximations thereof (e.g., rendering). Hence, our broad strategy to uncover the true underlying sources is to bias the model towards encoding to and decoding from a similarly structured latent space.

We manifest this inductive bias by drawing from two common ideas in the machine learning literature: discrete representations [55] and model regularization. Specifically, we propose (i) quantizing a model's latent representation into learnable discrete values with a separate scalar codebook per dimension and (ii) applying strong regularization via an unusually high weight decay [49]. Intuitively, forcing the model to use a small number of scalar values to combinatorially construct many latent codes encourages it to assign a consistent meaning to each value, an outcome that weight decay explicitly incentivizes by regularizing towards parsimony.

A side benefit of models with quantized latents is that they sidestep one issue that has hindered evaluation in previous works: they enable the use of simpler, more robust distribution estimation techniques for discrete variables. As a further methodological contribution, we present InfoMEC, a new set of metrics for the modularity, explicitness, and compactness of (both continuous and discrete) representations that is cohesively grounded in information theory and fixes other well-established shortcomings in existing disentanglement metrics.

We demonstrate the broad applicability of latent quantization by adding it to both basic data-reconstructing (vanilla autoencoder) and latent-reconstructing (InfoGAN) generative models. Together with regularization, this is sufficient to dramatically improve the modularity and explicitness of the learned representations of a representative suite of four disentangled representation learning datasets with image observations and ground-truth source evaluations [9, 20, 53]. In particular, our quantized-latent autoencoder (QLAE, pronounced like clay) consistently outperforms strong methods from prior work without compromising data reconstruction. We think of QLAE as a minimalist implementation of a combinatorial representation in neural networks, suggesting that our recipe of latent quantization and regularization could be broadly useful to other areas of machine learning.[1]

# 2 Preliminaries

In order to properly contextualize our proposed inductive bias, methodological contributions, and experiments, we first devote some attention to explaining the problem of disentangled representation learning. We also discuss prior disentangled representation learning methods we build upon.

## 2.1 Nonlinear ICA and Disentangled Representation Learning

We begin by considering the standard data-generating model of nonlinear independent components analysis (ICA) [30], a problem very related to but more conceptually precise than disentanglement:

$$p(\mathbf{s}) = \prod_{i=1}^{n_s} p(\mathbf{s}_i), \ \ \mathbf{x} = g(\mathbf{s}), \tag{1}$$

---

[1]Code for models and InfoMEC metrics: https://github.com/kylehkhsu/latent_quantization.

where $\mathbf{s} = (\mathbf{s}_1, \ldots, \mathbf{s}_{n_s})$ comprise the $n_s$ mutually independent source variables (sources); $\mathbf{x}$ is the observed data variable; and $g : \mathcal{S} \to \mathcal{X}$ is the nonlinear data-generating function. The nonlinear ICA problem is to recover the underlying sources given a dataset $\mathcal{D}$ of samples from this model. Specifically, the solution should include an approximate inverting function $\hat{g}^{-1} : \mathcal{X} \to \mathcal{Z}$ such that, assuming latent variables (latents) $\mathbf{z} = (\mathbf{z}_1, \ldots, \mathbf{z}_{n_s})$ are matched correctly to the sources, each source is perfectly determined by its corresponding latent. A typical technical phrasing is that $\hat{g}^{-1} \circ g$ should be the composition of a permutation and a dimension-wise invertible function.

As stated, this problem is *nonidentifiable* (or underspecified). Given $\mathcal{D}$, one may find many sets of independent latents (and associated nonlinear generators) that fit the data despite being non-trivially different from the true generative sources [30]. As such, reliably recovering the true sources from the data is impossible. Much recent work in nonlinear ICA has focused on proposing additional problem assumptions so as to provably pare down the possibilities to a unique solution. These theoretical assumptions can then be transcribed into architectural choices or regularization terms. Such approaches have shown promise in increasing our understanding of what assumptions are required to disentangle; we review these works in Section 6.

While identifiability is conceptually appealing, achieving it under sufficiently generalizable assumptions that apply to non-toy datasets has proven hard. The field of disentangled representation learning has taken a more pragmatic approach, focusing on empirically evaluating the recovery of each dataset's designated source set. Given this empirical focus, robust performance metrics are vital. Unfortunately, there is a plethora of approaches in use, and subtle yet impactful issues arise even in the most common choices [10]. To address these concerns, in Section 4 we propose new metrics for three existing, complementary notions of disentanglement [17, 59]. They measure the following three properties: **modularity**—the extent to which sources are encoded into disjoint sets of latents; **explicitness**—how *simply* the latents encode each source; and **compactness**—the extent to which latents encode information about disjoint sets of sources. We frame these three in a cohesive information-theoretic framework that we name InfoMEC.

The disentangled representation learning problem statement considered in this work is as follows. Given a dataset of paired source-data samples $\{(s, x = g(s))\}$ from the nonlinear ICA model (1), learn an encoder $\hat{g}^{-1} : \mathcal{X} \to \mathcal{Z}$ and decoder $\hat{g} : \mathcal{Z} \to \mathcal{X}$ solely using the data $\{x\}$ such that (i) the InfoMEC as estimated from samples $\{(s, z = \hat{g}^{-1} \circ g(s))\}$ from the joint source-latent distribution is high, while (ii) maintaining an acceptable level of reconstruction error between $x$ and $\hat{g} \circ \hat{g}^{-1}(x)$.

## 2.2 Autoencoding and InfoGAN as Data and Latent Reconstruction

We will apply our proposed latent quantization scheme to two foundational approaches for disentangled representation learning: vanilla autoencoders (AEs) and information-maximizing generative adversarial networks (InfoGANs) [12]. Here, we provide a brief overview of the two and defer complete implementation details to Appendices A and C. Both approaches involve learning an encoder $\hat{g}^{-1}$ and decoder $\hat{g}$. An autoencoder takes a datapoint $x \in \mathcal{X}$ as input and produces a reconstruction $\hat{g} \circ \hat{g}^{-1}(x) \in \mathcal{X}$ that is optimized to match the input:

$$\mathcal{L}_{\text{reconstruct data}}(\hat{g}^{-1}, \hat{g}; \mathcal{D}) := \mathbb{E}_{x \sim \mathcal{D}} \left[ -\log p(x \mid \hat{g} \circ \hat{g}^{-1}(x)) \right]. \tag{2}$$

An InfoGAN instead takes a latent code $z \in \mathcal{Z}$ as input. The decoder (aka generator) maps $z$ to the data space, and from this the encoder produces a reconstruction of the latent:

$$\mathcal{L}_{\text{reconstruct latent}}(\hat{g}^{-1}, \hat{g}) := \mathbb{E}_{z \sim p(\mathbf{z})} \left[ -\log p(z \mid \hat{g}^{-1} \circ \hat{g}(z)) \right]. \tag{3}$$

Unlike the data reconstruction loss, this is clearly insufficient for learning as the dataset $\mathcal{D}$ isn't even used. InfoGAN can be thought of as grounding latent reconstruction by making the marginal distribution of generated datapoints, $\hat{g}(\mathbf{z})$, indistinguishable from the empirical data distribution. A concrete measure of this is provided by an additional binary classifier (aka discriminator) [21] or value model (aka critic) [1] trained alongside but in opposition to the decoder. While InfoGAN was originally motivated as maximizing a variational lower bound on the mutual information between the latent and the generated data, we find the above interpretation to be unifying.

## 3 Latent Quantization

Our goal is to encourage our model to disentangle by biasing it towards using an organized latent space. Why would this mitigate the nonidentifiability of nonlinear ICA? Our key motivation is that

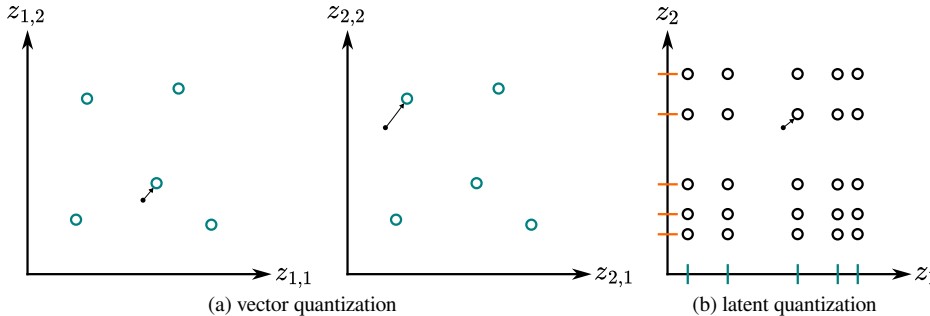

(a) vector quantization            (b) latent quantization

Figure 2: Two ways to quantize into one of $n_v{}^{n_z}$ ($5^2$) discrete codes. (a) Vector quantization [55] splits a continuous representation of size $n_z d$ (4) into $n_z$ parts (black dots), each of which is quantized to the nearest of a global codebook of $n_v$ vectors of size $d$ (turquoise circles). (b) Latent quantization specifies $d = 1$ and quantizes a continuous representation of size $n_z$ (black dot) onto a regular grid parameterized by dimension-specific codebooks (orange and turquoise ticks), each of size $n_v$. Unlike vector quantization, latent quantization ties the decoder input space to the discrete code space.

generative processes for realistic data are compositional and hence necessarily use highly organized source spaces. We discuss connections to related works in Section 6.

To build our desired inductive bias, we propose latent quantization, a modification of vector quantization [55]. In the latter, the latent representation of a datapoint is partitioned into component vectors, each of which is quantized to the nearest of a discrete set of vector embeddings (Figure 2a). We modify vector quantization to better enable the model to encode and decode with a consistent interpretation of each component of the discrete code. Concretely, we do this by specifying the set of latent codes to be the Cartesian product of $n_z$ distinct scalar codebooks: $Z = V_1 \times \cdots \times V_{n_z}$, where each codebook $V_j$ is a set of $n_v$ reals. The complete transformation comprises, first, the encoder network mapping the data to a continuous latent space $\hat{g}^{-1} : \mathcal{X} \to \mathbb{R}^{n_z}$ (with a slight abuse of notation), followed by quantization onto the nearest code (Figure 2b). This nearest neighbor calculation can be done elementwise, which is highly efficient. Formally, latent quantization is:

$$z_j = \underset{v_{jk} \in V_j}{\arg\min} \, |\hat{g}^{-1}(x)_j - v_{jk}|, \quad j = 1, \ldots, n_z. \tag{4}$$

We represent $Z$ by its constituent discrete values, concretely as a learnable two-dimensional array $V \in \mathbb{R}^{n_z \times n_v}$, the $j$-th row of which stores the elements of codebook $V_j$ in an arbitrary order.

A lesson from nonlinear ICA is that, given a flexible enough model, data can be mapped to and from latent spaces in many convoluted ways. We motivate the use of strong model regularization with the conjecture that, of all the possible mappings from organized latent space to data, the most parsimonious will be the true generative model or something close enough to it. We operationalize this by using a high weight decay on both the encoder and decoder networks. Ablation studies (Section 5) show that both quantized latents and weight decay are necessary to disentangle well.

To train a quantized-latent model, we use the straight-through gradient estimator [5] and co-opt the quantization and commitment losses proposed for vector quantization:

$$\mathcal{L}_{\text{quantize}} = \|\text{StopGradient}(\hat{g}^{-1}(x)) - z\|_2^2, \qquad \mathcal{L}_{\text{commit}} = \|\hat{g}^{-1}(x) - \text{StopGradient}(z)\|_2^2. \tag{5}$$

The straight-through gradient estimator facilitates the flow of gradients through the nondifferentiable quantization step. $\mathcal{L}_{\text{quantize}}$ pulls the discrete values constituting $z$ onesidedly towards the pre-quantized continuous output of the encoder. This is needed to optimize V, since straight-through gradient estimation disconnects V from the computation graph. Conversely, the commitment loss prevents the pre-quantized representation, which does see gradients from downstream computation, from straying too far from the codes. While this is a significant failure mode for vector quantization, we find that the use of scalars instead of high-dimensional vectors alleviates this issue, allowing us to drastically downweight the quantization and commitment losses while maintaining training stability. This gives the model much-needed flexibility to reorganize the discrete latent space. Finally, while using a shared global codebook like in vector quantization is certainly feasible, we find it better to maintain dimension-specific codebooks to enable the stable optimization of each individual value. Algorithm 1 contains pseudocode for latent quantization and computing the quantization and

commitment losses. Appendix A presents pseudocode for training a quantized-latent autoencoder (QLAE) in Algorithm 2 and a quantized-latent InfoWGAN-GP [12, 1, 23] in Algorithm 3.

---

**Algorithm 1** Latent quantization and computation of codebook losses.

---

1: **function** LatentQuantization(datapoint $x$, encoder $\hat{g}^{-1}$, discrete value array V)
2:      $z_c \leftarrow \hat{g}^{-1}(x)$
3:      $z \leftarrow \arg\min_{v \in Z} \|z_c - v\|_1, \ \ v_j \in V_j, \ \ \|_{j=1}^{n_z} V_j = \text{V}$      ▷ implement via (4)
4:      $\mathcal{L}_{\text{quantize}} \leftarrow \|\text{StopGradient}(z_c) - z\|_2^2$      ▷ from VQ-VAE [55]
5:      $\mathcal{L}_{\text{commit}} \leftarrow \|z_c - \text{StopGradient}(z)\|_2^2$      ▷ ibid.
6:      $z \leftarrow z_c + \text{StopGradient}(z - z_c)$      ▷ straight-through gradient estimator [5]
7:      **return** $z, \mathcal{L}_{\text{quantize}}, \mathcal{L}_{\text{commit}}$

---

# 4    InfoMEC: Information-Theoretic Metrics for Disentanglement

In this section, we derive InfoMEC, metrics for modularity, explicitness, and compactness, building upon and otherwise taking inspiration from several prior works [58, 17, 11, 34, 72, 45, 10]. We take care to motivate our design decisions, and while we do not expect this to be the final word on disentanglement metrics, we hope our presentation enables others to clearly understand InfoMEC and propose further improvements.

## 4.1    Modularity and Compactness

Nonlinear ICA asks for the latents to recover the sources up to a permutation and dimension-wise invertible transformation. The mutual information between an individual source and latent,

$$I(\mathbf{s}_i; \mathbf{z}_j) := D_{\text{KL}}(p(\mathbf{s}_i, \mathbf{z}_j) \parallel p(\mathbf{s}_i)p(\mathbf{z}_j)), \tag{6}$$

is a granular measure of the extent to which they are deterministic functions of each other. Unlike other measures such as correlation (used in MCC [34]), LASSO weights (used in linear DCI [17]), or linear predictive accuracy (used in SAP [41]), mutual information takes into account arbitrary nonlinear dependence between its two arguments, making it invariant within the nonlinear ICA equivalence class for any candidate solution.

When both arguments are discrete, estimating the mutual information is simple via the empirical joint distribution, but if either is continuous, estimation becomes non-trivial. Previous works bin a continuous variable and pretend it is discrete [47, 10], but this is sensitive to the binning strategy [10]. Instead, for evaluating continuous latents, we choose the celebrated $k$-nearest neighbor based KSG estimator [40, 19], in particular a variant [60] designed to handle a mix of discrete and continuous arguments. We use $k = 3$. See Appendix B for experimental vignettes demonstrating the severe sensitivity of binning-based estimation to the binning strategy (Figure 5) and the robustness of KSG-based estimation to $k$ (Figure 6). We remark that latent quantization enables reliable evaluation using the discrete-discrete estimator.

To facilitate aggregation, we desire a normalization to the interval $[0, 1]$. To this end, note that the identity $I(\mathbf{s}_i; \mathbf{z}_j) = H(\mathbf{s}_i) - H(\mathbf{s}_i \mid \mathbf{z}_j)$ and the nonnegativity of entropy imply $I(\mathbf{s}_i; \mathbf{z}_j) \leq H(\mathbf{s}_i)$ for discrete sources. Following [11], we define a normalized mutual information as

$$\text{NMI}(\mathbf{s}_i, \mathbf{z}_j) := \frac{I(\mathbf{s}_i; \mathbf{z}_j)}{H(\mathbf{s}_i)}. \tag{7}$$

We prefer this normalization scheme over others [66] since i) it is the proportion of a source's entropy reduced by conditioning on a latent and thus scales consistently to $[0, 1]$ for any model, and ii) it avoids the scale-dependent (and possibly negative) differential entropy of a continuous latent. We gather all evaluations of $\text{NMI}(\mathbf{s}_i, \mathbf{z}_j)$ into a 2-dimensional array $\text{NMI} \in [0, 1]^{n_s \times n_z}$. We remove inactive latents (columns of NMI), which are those with zero range (over the evaluation sample) for discrete latents. For continuous latents, zero is too strict, so we heuristically define the threshold to be $1/20$, applied after dividing the ranges by their maximum. See Figure 3 for examples of $\text{NMI}^\top$.

Modularity is the extent to which sources are separated into disjoint sets of latents. Perfect modularity occurs when each latent is informative of only one source, i.e. when every column of NMI has only one nonzero element. This has been measured as the *gap* [11] between the two largest entries in a

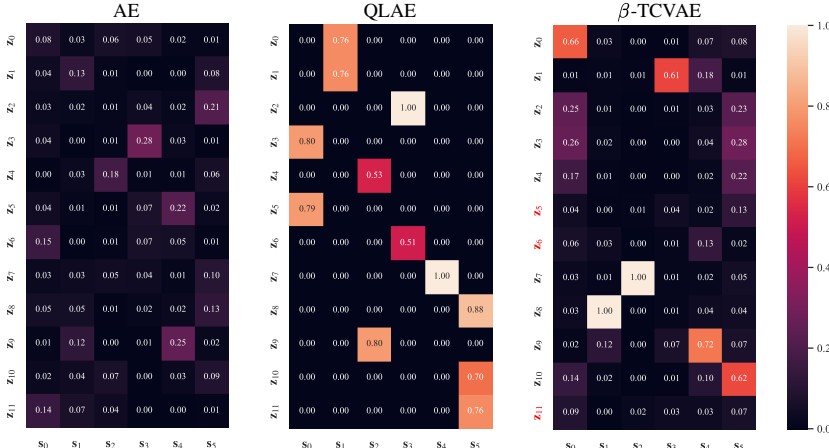

Figure 3: Visualization of $\mathrm{NMI}^\top$ for Shapes3D models. Inactive latents (red font) are removed from InfoM and InfoC computation. The low values in the $\mathrm{NMI}^\top$ of AE indicate that individual latents are not highly informative of individual sources. Adding latent quantization results in QLAE achieving near-perfect InfoM (0.99): each latent is only informative of one source (rows are sparse). This induces a trade-off with InfoC, in which $\beta$-TCVAE scores higher (0.53 vs. 0.62). See Appendix D for qualitative studies on the fidelity of NMI entries as judged by decoded latent interventions.

column, or the *ratio* [72] of the largest entry in the column to the column sum. We prefer the ratio since the gap is agnostic to the smallest $n_s - 2$ values in the column, but these values matter and should influence the measure [10]. Since the possible range of values for this ratio is $[1/n_s, 1]$, we re-normalize to $[0, 1]$. Finally, we define InfoModularity (InfoM) as the average of this quantity over latents:

$$\mathrm{InfoM} := \left( \frac{1}{n_z} \sum_{j=1}^{n_z} \frac{\max_i \mathrm{NMI}_{ij}}{\sum_{i=1}^{n_s} \mathrm{NMI}_{ij}} - \frac{1}{n_s} \right) \Big/ \left( 1 - \frac{1}{n_s} \right). \tag{8}$$

Compactness complements modularity; it is the extent to which latents only contain information about disjoint sets of sources. We therefore define InfoCompactness (InfoC) analogously to InfoM, but considering rows of NMI instead of columns, and averaging over sources instead of latents, etc.:

$$\mathrm{InfoC} := \left( \frac{1}{n_s} \sum_{i=1}^{n_s} \frac{\max_j \mathrm{NMI}_{ij}}{\sum_{j=1}^{n_z} \mathrm{NMI}_{ij}} - \frac{1}{n_z} \right) \Big/ \left( 1 - \frac{1}{n_z} \right). \tag{9}$$

We advocate for this terminology since previous names such as "mutual information gap" [11] and "mutual information ratio" [72] are ambiguous, and indeed the former of these works considered solely compactness and the latter solely modularity, with neither mentioning the distinction. We remark that when $n_z > n_s$ (after pruning inactive latents), it is impossible to achieve both perfect modularity and perfect compactness. Of the two, modularity should be prioritized [58, 10] and indeed has been referred to as disentanglement itself [17].

## 4.2 Explicitness

Modularity and compactness are measured in terms of mutual information, so they are agnostic to *how* this information is encoded. Our third metric, explicitness, measures the extent to which the relationship between the sources and latents is simple (e.g., linear [41, 17, 59, 18]). Since previous explicitness metrics have been rather ad hoc, we propose a formalism using the framework of predictive $\mathcal{V}$-information, a generalization of mutual information that specifies an allowable function class, denoted $\mathcal{V}$, for the computation of information [75]. We first estimate the predictive $\mathcal{V}$-information of each source $\mathbf{s}_i$ given *all* latents $\mathbf{z}$:

$$I_\mathcal{V}(\mathbf{z} \to \mathbf{s}_i) := H_\mathcal{V}(\mathbf{s}_i \mid \varnothing) - H_\mathcal{V}(\mathbf{s}_i \mid \mathbf{z}). \tag{10}$$

This requires estimating the predictive conditional $\mathcal{V}$-entropy

$$H_\mathcal{V}(\mathbf{s}_i \mid \mathbf{z}) := \inf_{f \in \mathcal{V}} \mathbb{E}_{s \sim p(\mathbf{s}), z \sim p(\mathbf{z} \mid \mathbf{s})} \left[ -\log p(s_i \mid f(z)) \right] \tag{11}$$

and the marginal $\mathcal{V}$-entropy of the source

$$H_\mathcal{V}(\mathbf{s}_i \mid \varnothing) := \inf_{f \in \mathcal{V}} \mathbb{E}_{s \sim p(\mathbf{s})} \left[ -\log p(s_i \mid f(\varnothing)) \right], \tag{12}$$

where $\varnothing$ is an uninformative constant. The predictive conditional $\mathcal{V}$-entropy measures how well a source, $\mathbf{s}_i$, can be predicted by mapping the latents, $\mathbf{z}$, through a function in function class $\mathcal{V}$. Note that this estimation uses the best *in-sample* negative log likelihood. We choose $\mathcal{V}$ to be the space of linear models (though one could pick $\mathcal{V}$ to fit particular needs) and so use logistic regression (linear regression) for discrete (continuous) sources. We use no regularization. We compute the marginal $\mathcal{V}$-entropy $H_{\mathcal{V}}(\mathbf{s}_i \mid \varnothing)$ in the same way, but substituting a universal constant for all inputs. We propose a simple normalization analogous to the one done for NMI:

$$\mathrm{NMI}_{\mathcal{V}}(\mathbf{z} \to \mathbf{s}_i) := \frac{I_{\mathcal{V}}(\mathbf{z} \to \mathbf{s}_i)}{H_{\mathcal{V}}(\mathbf{s}_i \mid \varnothing)}, \tag{13}$$

which can be interpreted as the relative reduction in the $\mathcal{V}$-entropy of a source achieved by knowing the latents, and is in $[0, 1]$: for classification negative log likelihood is the cross-entropy, and for regression we leverage Propositions 1.3 and 1.5 from Xu et al. [75] to argue that $\mathrm{NMI}_{\mathcal{V}}(\mathbf{z} \to \mathbf{s}_i) = R^2$, the coefficient of determination. We can now compute explicitness as:

$$\mathrm{InfoE} := \frac{1}{n_s} \sum_{i=1}^{n_s} \mathrm{NMI}_{\mathcal{V}}(\mathbf{z} \to \mathbf{s}_i). \tag{14}$$

### 4.3 Summary and Comparison to Nonlinear DCI

We have derived three metrics for evaluating the modularity, explicitness, and compactness of a representation. Each metric has a straightforward information-theoretic interpretation and all share a range of $[0, 1]$. We order them in decreasing importance and collectively refer to them as $\mathrm{InfoMEC} := (\mathrm{InfoM}, \mathrm{InfoE}, \mathrm{InfoC})$.

Nonlinear DCI [17], a widely used three-pronged framework that measures similar disentanglement properties, suffers several practical drawbacks in comparison to InfoMEC from being defined in terms of relative counts of decision tree splits: determining this requires considering all latents *jointly* while fitting $p(\mathbf{s}_i \mid \mathbf{z})$. This results in a cumbersome computational footprint that is exacerbated by highly sensitive hyperparameters such as tree depth [17, 10], the tuning of which has even seen omission in prior work [47]. These drawbacks worsen with increased latent space dimensionality. In contrast, InfoMEC avoids these issues as it isolates InfoM and InfoC from the choice of predictive function class and only computes pairwise interactions between individual sources and latents. (InfoE also fits $p(\mathbf{s}_i \mid \mathbf{z})$, but does so with function classes of severely limited capacity for which fitting procedures scale well.) See Appendix B for experimental vignettes demonstrating the hyperparameter sensitivity of nonlinear DCI (Figure 7) and the robustness of InfoMEC (Figure 6).

## 5 Experiments

**Experimental design.** We design our experiments to answer the following questions: Does latent quantization improve disentanglement? How does it compare against the strongest known methods that operate under the same assumptions? And, finally, which of our design choices were critical? We benchmark on four established datasets: Shapes3D [9], MPI3D [20], Falcor3D [53], and Isaac3D [53]. Each consists of RGB image observations generated (near-)noiselessly from categorical or discretized numerical sources. Shapes3D is toyish, but the others are chosen for their difficulty [54, 20]. In particular, we use the `complex shapes` variant of MPI3D collected on a real world robotics apparatus. See Appendix C.1 for further dataset details. Aside from baseline AE and InfoGAN (specifically InfoWGAN-GP, a Wasserstein GAN [1] with gradient penalty [23]), we compare to $\beta$-VAE [25], $\beta$-TCVAE [11], and BioAE [72], the strongest methods from prior work that obey our problem assumptions and make design decisions mutually exclusive with latent quantization. We also compare to VQ-VAE with $d = 64$ and $n_v = 512$ (Figure 2). We ablate weight decay, scalar codebooks, and dimension-specific codebooks from QLAE and weight decay from QLInfoWGAN-GP. We quantify modularity, explicitness, and compactness using both InfoMEC and nonlinear DCI. We qualitatively inspect representations via decoded latent interventions (Figure 4 and Appendix D).

**Select experimental details.** The choice of decoder architecture is known to impose inductive biases relevant for disentanglement [33, 43]. We use an expressive architecture (Appendix C.3) based on StyleGAN [33, 54] for all methods and datasets. We downsample the observations to $64 \times 64$ (if necessary). We follow prior work [47] in considering a statistical learning problem rather than a machine learning one: we train on the entire dataset then evaluate on $10,000$ i.i.d. samples. We fix

the number of latents in all methods to twice the number of sources. For quantized-latent models, we fix $n_v = 10$ discrete values per codebook. We tune one key regularization hyperparameter per method per dataset with a thorough sweep (Table 11, Appendix C.2). We use the best performing configurations over 2 seeds and rerun with 5 more seeds. Despite our modest list of methods and ablations, just the last stage took over 1000 GPU-hours.

Table 1: Main disentanglement results measured in InfoMEC and nonlinear DCI. Modularity is the key property, followed by explicitness, with compactness (grayed) a distant third. AE and InfoGAN variants are presented and bolded separately as all AEs are filtered for near-perfect data reconstruction, whereas InfoGANs are generally more lossy. For confidence intervals and data reconstruction results, see Appendix E.

| model | aggregated | Shapes3D | MPI3D | Falcor3D | Isaac3D |
|---|---|---|---|---|---|
| | InfoMEC := (InfoM InfoE InfoC) ↑ | | | | |
| AE | (0.40 **0.81** 0.26) | (0.41 **0.98** 0.28) | (0.37 **0.72** 0.36) | (0.39 0.74 0.20) | (0.42 0.80 0.21) |
| β-VAE | (0.59 0.81 0.55) | (0.59 **0.99** 0.49) | (0.45 **0.71** 0.51) | (**0.71** 0.73 0.70) | (0.60 0.80 0.51) |
| β-TCVAE | (0.58 0.72 0.59) | (0.61 0.82 0.62) | (0.51 0.60 0.57) | (**0.66** 0.74 0.71) | (0.54 0.70 0.46) |
| BioAE | (0.54 0.75 0.36) | (0.56 **0.98** 0.44) | (0.45 **0.66** 0.36) | (0.54 0.73 0.31) | (0.63 0.65 0.33) |
| VQ-VAE | (0.58 **0.81** 0.39) | (0.72 **0.97** 0.47) | (0.43 0.57 0.22) | (0.61 **0.83** 0.42) | (0.57 0.87 0.45) |
| QLAE (ours) | (**0.76 0.84** 0.50) | (**0.95 0.99** 0.55) | (**0.61** 0.63 0.51) | (**0.71 0.77** 0.44) | (**0.78 0.97** 0.49) |
| InfoWGAN-GP | (0.50 **0.57** 0.29) | (0.61 **0.78** 0.41) | (0.43 0.40 0.20) | (0.44 **0.60** 0.30) | (0.53 0.51 0.24) |
| QLInfoWGAN-GP (ours) | (**0.63 0.59** 0.47) | (**0.73 0.75** 0.48) | (**0.62 0.51** 0.37) | (**0.54 0.53 0.56**) | (**0.63 0.58** 0.49) |
| | DCI := (D I C) ↑ | | | | |
| AE | (0.12 0.81 0.10) | (0.11 0.82 0.08) | (0.15 **0.83** 0.14) | (0.08 0.76 0.07) | (0.13 0.85 0.10) |
| β-VAE | (0.37 0.89 0.30) | (0.61 0.99 0.47) | (**0.31 0.83** 0.27) | (0.32 0.84 0.28) | (0.23 0.88 0.19) |
| β-TCVAE | (0.31 0.87 0.27) | (0.46 **0.99** 0.38) | (0.22 0.77 0.21) | (0.36 0.90 0.33) | (0.19 0.84 0.16) |
| BioAE | (0.29 0.86 0.23) | (0.33 0.93 0.25) | (0.24 **0.79** 0.19) | (0.21 0.81 0.17) | (0.38 0.91 0.31) |
| VQ-VAE | (0.28 0.79 0.27) | (0.40 0.84 0.34) | (0.09 0.63 0.14) | (0.30 0.79 0.29) | (0.33 0.89 0.31) |
| QLAE (ours) | (**0.59 0.95** 0.47) | (**0.81 0.99** 0.61) | (**0.36 0.85** 0.36) | (**0.50 0.96** 0.38) | (**0.69 0.99** 0.54) |
| InfoWGAN-GP | (0.14 0.72 0.12) | (0.23 0.80 0.18) | (0.09 0.63 0.09) | (0.11 **0.74** 0.08) | (0.13 0.71 0.11) |
| QLInfoWGAN-GP (ours) | (**0.26 0.77** 0.26) | (**0.38 0.85** 0.29) | (**0.24 0.71** 0.25) | (**0.20 0.73** 0.24) | (**0.24 0.79** 0.25) |

**Effect of latent quantization.** Adding latent quantization to AE and InfoWGAN-GP results in consistent and dramatic increases in modularity and compactness under both InfoMEC and DCI evaluation (Table 1). For explicitness, the improvement is more significant under (random forest) I than under (linear) InfoE. Qualitatively, decoded latent interventions demonstrate that the QLAE latent space is highly interpretable, corroborating the gains in modularity (Figure 4 and Appendix D).

**Comparison of QLAE with prior methods.** QLAE significantly outperforms all prior methods on all four datasets in modularity under both InfoM and D (Table 1), with the exception of β-VAE and β-TCVAE on Falcor3D under InfoM. There is also significant improvement in explicitness under (random forest) I, but less so for (linear) InfoE. The objectives in β-VAE and β-TCVAE contain a term that explicitly minimizes the total correlation (aka multiinformation), amongst the latent variables [11]. Since this essentially optimizes for compactness, we should expect compactness metrics to rank β-VAE and β-TCVAE ahead of QLAE; InfoC does so, but C does not. Corresponding metrics from InfoMEC and DCI have Spearman rank correlations of $\rho = 0.80, p < 1 \times 10^{-11}$ for modularity, $\rho = 0.77, p < 1 \times 10^{-9}$ for explicitness, and $\rho = 0.59, p < 1 \times 10^{-5}$ for compactness.

Table 2: Ablation studies on QLAE and QLInfoWGAN-GP.

| model | aggregated | Shapes3D | MPI3D | Falcor3D | Isaac3D |
|---|---|---|---|---|---|
| | InfoMEC := (InfoM InfoE InfoC) ↑ | | | | |
| QLAE (ours) | (**0.76 0.84** 0.50) | (**0.95 0.99** 0.55) | (**0.61** 0.63 0.51) | (**0.71** 0.77 0.44) | (**0.78 0.97** 0.49) |
| QLAE w/ global codebook | (0.68 0.80 0.44) | (**0.96 0.99** 0.48) | (0.54 0.62 0.45) | (0.59 0.74 0.37) | (0.65 0.82 0.46) |
| QLAE w/o weight decay | (0.63 0.81 0.48) | (0.69 **0.99** 0.51) | (0.51 0.61 0.48) | (0.65 0.76 0.43) | (0.66 0.89 0.51) |
| VQ-VAE w/ weight decay | (0.69 **0.87** 0.43) | (0.80 **0.99** 0.46) | (0.50 **0.81** 0.41) | (**0.74 0.86** 0.40) | (**0.73** 0.81 0.44) |
| QLInfoWGAN-GP (ours) | (**0.63 0.59** 0.47) | (**0.73 0.75** 0.48) | (**0.62 0.51** 0.37) | (**0.54 0.53 0.56**) | (**0.63 0.58** 0.49) |
| QLInfoWGAN-GP w/o w.d. | (**0.61** 0.55 0.44) | (0.66 0.57 0.40) | (**0.63 0.55** 0.40) | (**0.56 0.56** 0.52) | (**0.59 0.53** 0.45) |
| | DCI := (D I C) ↑ | | | | |
| QLAE (ours) | (**0.59 0.95** 0.47) | (**0.81 0.99** 0.61) | (**0.36 0.85** 0.36) | (**0.50 0.96** 0.38) | (**0.69 0.99** 0.54) |
| QLAE w/ global codebook | (0.52 **0.93** 0.41) | (**0.83 0.99** 0.60) | (**0.36 0.86** 0.34) | (0.36 0.90 0.26) | (0.53 **0.99** 0.43) |
| QLAE w/o weight decay | (0.49 **0.94** 0.40) | (0.63 **0.99** 0.47) | (0.30 **0.84** 0.29) | (0.46 **0.97** 0.36) | (0.58 **0.99** 0.48) |
| VQ-VAE w/ weight decay | (0.43 0.84 0.37) | (0.74 **0.99** 0.57) | (0.22 0.68 0.20) | (0.41 0.85 0.32) | (0.34 0.85 0.39) |
| QLInfoWGAN-GP (ours) | (**0.26 0.77 0.26**) | (**0.38 0.85** 0.29) | (**0.24 0.71** 0.25) | (**0.20 0.73** 0.24) | (**0.24 0.79** 0.25) |
| QLInfoWGAN-GP w/o w.d. | (0.19 0.73 0.19) | (0.16 0.71 0.13) | (**0.28 0.74** 0.23) | (0.14 **0.72** 0.17) | (**0.20 0.77** 0.23) |

**Ablations on latent space design and regularization.** We observe that ablating dimension-specific codebooks, weight decay, and scalar codebooks from QLAE each causes a significant drop in InfoM and D (Table 2), verifying the importance of these design decisions. The effect of ablating weight decay from QLInfoWGAN-GP is less pronounced. We note that while VQ-VAE w/ weight decay performs somewhat closely to QLAE in terms of InfoMEC, this is only because we use the categorical codes (as opposed to the high-dimensional vector representation) for evaluation. In addition, the scalar codebook design of QLAE enables meaningful interpolation between discrete values, whereas this is not supported by vector quantization.

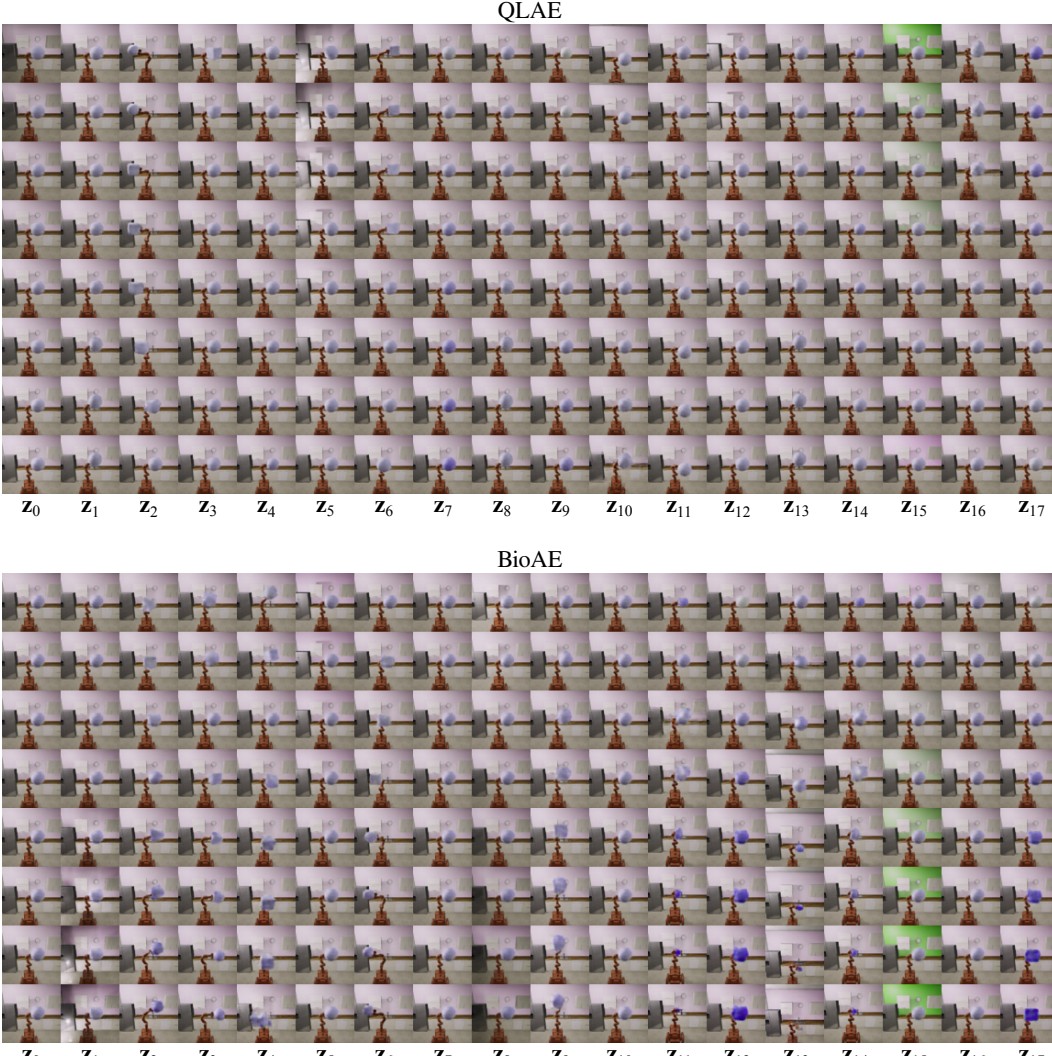

Figure 4: Decoded Isaac3D latent interventions for QLAE and BioAE, the prior method with highest InfoM. In each image block, a single data sample is encoded into its latent representation. In the $j$-th column, the $j$-th latent is intervened on with a linear interpolation of its range across the dataset. The resulting representation is then decoded. For reference, the true sources for this dataset are object shape, robot $x$, robot $y$, camera height, object scale, lighting intensity, lighting direction, object color, and wall color. QLAE's representation is highly interpretable: going down each column corresponds to one source (sometimes two) changing in a consistent manner, with all other sources remaining unchanged. In contrast, the BioAE's representation varies more sporadically, and its decoder often generates low-quality samples from post-intervention latents. For qualitative results for multiple data samples from all datasets alongside NMI heatmaps, see Appendix D.

# 6 Related Work

**Nonlinear ICA and disentangled representation learning.** There is a long history of trying to build interpretable representations that separately represent the sources of variation in a dataset. This goes back to classic work on (linear) ICA [13, 29], and has been known in deep learning as disentanglement [4]. Without further assumptions, nonlinear ICA and its relatives in disentangled representation learning are provably underspecified [30, 47]. Approaches to resolve this indeterminacy include labeling a small number of datapoints [54] or showing pairs of datapoints in which only one or a few sources differ [63, 48]. More in line with our work are approaches that assume additional structure in the data generative process and disentangle by ensuring the representation reflects this structure [74]. Assumptions and methods include: factorized priors [25, 11, 36, 61], biologically inspired activity constraints [72], sparse source variation over time [65, 38], structurally sparse source to pixel influence [57, 31, 78, 52, 8], geometric assumptions on the source to image mapping [64, 77, 26, 22], sparse underlying causal graphs between sources [42], piecewise linearity [37], and hierarchical generation [43]. Many of these ideas are in principle compatible with latent quantization, and we leave discovery of fruitful combinations to future work.

**Factorized latent spaces.** VAEs that specify an isotropic Gaussian latent prior regularize the marginalized variational distribution (aka aggregate posterior) towards being a factorized distribution [11]. Roth et al. [61] bias latents to have pairwise factorized support via a Hausdorff set distance regularization. For linear ICA, Whittington et al. [72] prove that regularizing latents to be nonnegative and energy-minimizing results in them having factorized support. Differently from all of these works, latent quantization imbues a model with factorized structure by construction, instead of relying on the optimization of regularized objectives to manifest this structure. The favorable disentanglement that QLAE yields over $\beta$-TCVAE [11] and BioAE [72] suggests that this strategy is more effective.

**Discrete representation learning.** Oord et al. [55] first demonstrated the feasibility of discrete neural representation learning at scale, and their techniques have since been broadly applied, e.g., to videos [70, 76], audio [2, 15, 68, 6], and anomaly detection [50]. The following works design discrete representations similarly to how we do, though for purposes other than unsupervised disentanglement. Several works use one scalar codebook per latent dimension to achieve high efficiency in retrieval [3, 67, 32, 73]. Kobayashi et al. [39] disentangle normal and abnormal features in medical images into separate vector codebooks via pixel-space supervision. Liu et al. [46] and Träuble et al. [69] use multiple codebooks with separately parameterized key and value vectors and investigate the effect of discretization in systematic generalization and continual learning, respectively.

# 7 Discussion

We have proposed to use latent quantization and model regularization to impose an inductive bias towards disentanglement that enables our models to outperform strong prior methods. Ablations verify that our main design decisions are critical. We have also synthesized previously proposed ideas for evaluation into InfoMEC, three information-theoretic disentanglement metrics that rectify or sidestep key drawbacks in existing approaches.

While our results are promising, one concern might be that we have overfit our inductive bias to existing disentanglement benchmarks, in which, just like our model, the sources are discrete and the generative process is (near-)noiseless. Our experiments have already demonstrated the ability of latent quantization to represent sources that have more values (up to 40) than the per-dimension codebook size (fixed to 10) via allocating multiple latent dimensions. Future work should strive to construct disentanglement benchmarks that better reflect realistic conditions, e.g. continuous sources.

Beyond the intuitions and connections to related works we have presented, we do not understand why our method performs as well as it does. It may be fruitful to tackle this empirically, e.g. by probing how QLAE distributes data around its latent space, and how weight decay changes this. Achieving satisfactory understanding would enable the field to better position latent quantization within the ongoing body of work that aims to develop generalizable conditions for successful disentanglement.

Lastly, we hope this method, its future versions, and other methods the field develops are able to deliver on the original motivation for disentangled representation learning—to learn human-interpretable representations in complex, real-world situations, and to leverage the interpretability to empower human decision-making. This will require methods that can disentangle out-of-distribution data samples, that work for generic data types, and that can learn compositionally from sparse interactions with data. We suspect that latent quantization may have a role to play in these directions.

## Acknowledgments and Disclosure of Funding

We gratefully acknowledge the developers of open-source software packages that facilitated this research: NumPy [24], JAX [7], Equinox [35], matplotlib [28], seaborn [71], and scikit-learn [56]. We also thank Evan Liu, Kaylee Burns, Karsten Roth, Anirudh Goyal, and Cian Eastwood for feedback on previous drafts.

KH was funded by a Sequoia Capital Stanford Graduate Fellowship. Part of WD's work on this project happened during a visit to Stanford funded by the Bogue Fellowship. JCRW was funded by a Henry Wellcome Post-doctoral Fellowship (222817/Z/21/Z). This work was also in part supported by the Stanford Institute for Human-Centered AI (HAI), NSF RI #2211258, Air Force Office of Scientific Research (AFOSR) YIP FA9550-23-1-0127, and ONR MURI N00014-22-1-2740.

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

# A  Quantized-Latent Models

This section contains pseudocode for latent quantization and for training QLAE and QLInfoWGAN-GP.

---

**Algorithm 2** Pseudocode for optimizing a quantized-latent autoencoder (QLAE).
Hyperparameter values are detailed in Appendix C.2.

---

**Require:** dataset $\mathcal{D}$, batch size $b$, AdamW hyperparameters $(\alpha, \beta_1, \beta_2, \text{weight decay})$,
loss weights $\lambda := (\lambda_{\text{reconstruct data}}, \lambda_{\text{quantize}}, \lambda_{\text{commit}})$
1: initialize encoder $\hat{g}^{-1} : \mathcal{X} \to \mathbb{R}^{n_z}$, discrete value array $V \in \mathbb{R}^{n_z \times n_v}$, decoder $\hat{g} : \mathbb{R}^{n_z} \to \mathcal{X}$
2: **while** $(\hat{g}^{-1}, V, \hat{g})$ has not converged **do**
3:     **for** $i = 1, \ldots, b$ **do**
4:         $x \sim \mathcal{D}$
5:         $z, \mathcal{L}_{\text{quantize}}, \mathcal{L}_{\text{commit}} \leftarrow \text{LatentQuantization}(x, \hat{g}^{-1}, V)$            $\triangleright$ Algorithm 1
6:         $\mathcal{L}_{\text{reconstruct data}} \leftarrow \text{BinaryCrossEntropy}(\hat{g}(z), x)$
7:         $\mathcal{L}_{\text{QLAE}}^{(i)} \leftarrow \lambda \cdot (\mathcal{L}_{\text{reconstruct data}}, \mathcal{L}_{\text{quantize}}, \mathcal{L}_{\text{commit}})$
8:     $(\hat{g}^{-1}, \hat{g}) \leftarrow \text{AdamW}(\nabla_{(\hat{g}^{-1}, \hat{g})} \frac{1}{b} \sum_{i=1}^{b} \mathcal{L}_{\text{QLAE}}^{(i)}, (\hat{g}^{-1}, \hat{g}), \alpha, \beta_1, \beta_2, \text{weight decay})$
9:     $V \leftarrow \text{Adam}(\nabla_V \frac{1}{b} \sum_{i=1}^{b} \mathcal{L}_{\text{QLAE}}^{(i)}, V, \alpha, \beta_1, \beta_2)$

---

**Algorithm 3** Pseudocode for optimizing a quantized-latent InfoWGAN-GP.
Hyperparameter values are detailed in Appendix C.2.

---

**Require:** dataset $\mathcal{D}$, batch size $b$, AdamW hyperparameters $(\alpha, \beta_1, \beta_2, \text{weight decay})$,
encoder-to-generator update ratio $n_{e:g}$
loss weights $\lambda := (\lambda_{\text{reconstruct latent}}, \lambda_{\text{quantize}}, \lambda_{\text{commit}}, \lambda_{\text{value}}, \lambda_{\text{gradient penalty}})$
1: initialize encoder $\hat{g}^{-1} : \mathcal{X} \to \mathbb{R}^{n_z}$, discrete value array $V \in \mathbb{R}^{n_z \times n_v}$, decoder $\hat{g} : \mathbb{R}^{n_z} \to \mathcal{X}$
2: initialize critic $\hat{g}_c^{-1} : \mathcal{X} \to \mathbb{R}$
3: **while** $\theta := (\hat{g}^{-1}, V, \hat{g}, \hat{g}_c^{-1})$ has not converged **do**
4:     **for** $t = 1, \ldots, n_{e:g}$ **do**            $\triangleright$ encoder and critic updates
5:         **for** $i = 1, \ldots, b$ **do**
6:             $x_{\text{real}} \sim \mathcal{D}, \ z_{\text{fake}} \sim Z, \ \epsilon \sim \text{Uniform}([0, 1])$
7:             $x_{\text{fake}} \leftarrow \hat{g}(z_{\text{fake}})$
8:             $\mathcal{L}_{\text{value}} \leftarrow \hat{g}_c^{-1}(x_{\text{fake}}) - \hat{g}_c^{-1}(x_{\text{real}})$            $\triangleright$ from WGAN [1]
9:             $x_{\text{interpolated}} \leftarrow \epsilon x_{\text{real}} + (1 - \epsilon) x_{\text{fake}}$
10:           $\mathcal{L}_{\text{gradient penalty}} \leftarrow (\|\nabla_{x_{\text{interpolated}}} \hat{g}_c^{-1}(x_{\text{interpolated}})\|_2 - 1)^2$      $\triangleright$ from WGAN-GP [23]
11:           $\hat{z}_{\text{fake}}, \mathcal{L}_{\text{quantize}}, \mathcal{L}_{\text{commit}} \leftarrow \text{LatentQuantization}(x_{\text{fake}}, \hat{g}^{-1}, V)$     $\triangleright$ Algorithm 1
12:           $\mathcal{L}_{\text{reconstruct latent}} \leftarrow \text{MeanSquaredError}(\hat{z}_{\text{fake}}, z_{\text{fake}})$       $\triangleright$ from InfoGAN [12]
13:           $\mathcal{L}^{(i)} \leftarrow \lambda \cdot (\mathcal{L}_{\text{reconstruct latent}}, \mathcal{L}_{\text{quantize}}, \mathcal{L}_{\text{commit}}, \mathcal{L}_{\text{value}}, \mathcal{L}_{\text{gradient penalty}})$
14:         $(\hat{g}^{-1}, \hat{g}_c^{-1}) \leftarrow \text{AdamW}(\nabla_{(\hat{g}^{-1}, \hat{g}_c^{-1})} \frac{1}{b} \sum_{i=1}^{b} \mathcal{L}^{(i)}, (\hat{g}^{-1}, \hat{g}_c^{-1}), \alpha, \beta_1, \beta_2, \text{weight decay})$
15:     **for** $i = 1, \ldots, b$ **do**
16:         $z_{\text{fake}} \sim Z$
17:         $x_{\text{fake}} \leftarrow \hat{g}(z_{\text{fake}})$
18:         $\mathcal{L}_{\text{value}} \leftarrow -\hat{g}_c^{-1}(x_{\text{fake}})$            $\triangleright$ from WGAN [1]
19:         $\hat{z}_{\text{fake}}, \mathcal{L}_{\text{quantize}}, \mathcal{L}_{\text{commit}} \leftarrow \text{LatentQuantization}(x_{\text{fake}}, \hat{g}^{-1}, V)$     $\triangleright$ Algorithm 1
20:         $\mathcal{L}_{\text{reconstruct latent}} \leftarrow \text{MeanSquaredError}(\hat{z}_{\text{fake}}, z_{\text{fake}})$       $\triangleright$ from InfoGAN [12]
21:         $\mathcal{L}^{(i)} \leftarrow \lambda \cdot (\mathcal{L}_{\text{reconstruct latent}}, \mathcal{L}_{\text{quantize}}, \mathcal{L}_{\text{commit}}, \mathcal{L}_{\text{value}}, 0)$
22:     $\hat{g} \leftarrow \text{AdamW}(\nabla_{\hat{g}} \frac{1}{b} \sum_{i=1}^{b} \mathcal{L}^{(i)}, \hat{g}, \alpha, \beta_1, \beta_2, \text{weight decay})$
23:     $V \leftarrow \text{Adam}(\nabla_V \frac{1}{b} \sum_{i=1}^{b} \mathcal{L}^{(i)}, V, \alpha, \beta_1, \beta_2)$

# B    Disentanglement Metrics Vignettes

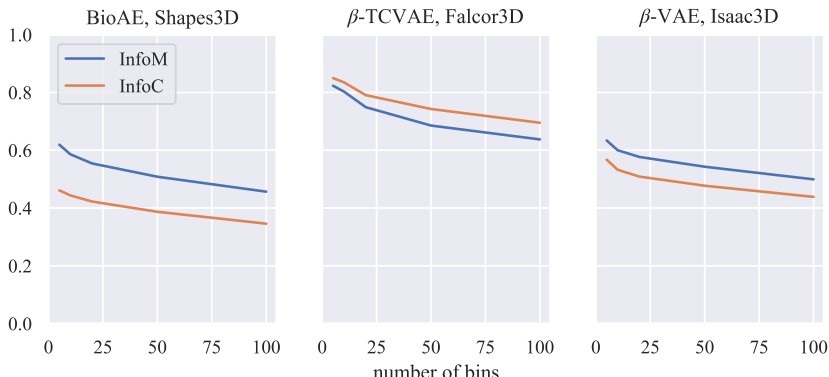

Figure 5: InfoM and InfoC for three models with continuous latents computed from mutual information estimated via histogram binning. The binning strategy has a drastic effect on the metrics, and it is unclear how it should be chosen.

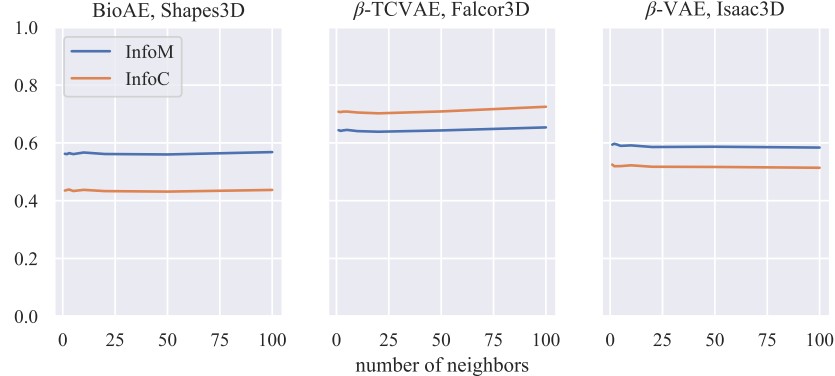

Figure 6: InfoM and InfoC for three models with continuous latents computed from mutual information estimated via $k$-neighbors based KSG [60]. The metrics are very robust to the choice of $k$.

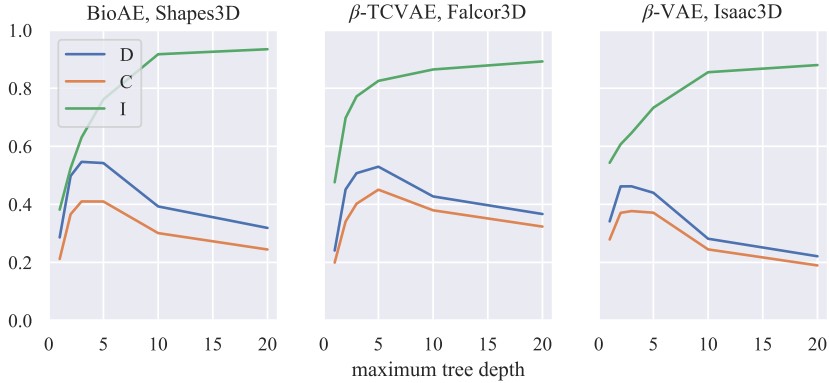

Figure 7: Nonlinear DCI for three models with continuous latents computed via random forests of 100 trees. D and C are as defined in Eastwood and Williams [17]. I is validation accuracy. DCI is highly sensitive to the maximum tree depth hyperparameter, and it is easy to overestimate D and C if this is not tuned with respect to I.

# C  Experiment Details

This section contains details on the experiments conducted in this work.

## C.1  Datasets

Table 3: Summary of datasets used for empirical evaluation.

| dataset | $n_s$ | $|\mathcal{D}|$ | description |
|---|---|---|---|
| Shapes3D [9] | 6 | 480,000 | geometric shapes in a scene with simplistic textures |
| MPI3D [20] | 7 | 460,800 | real robot arm holding objects in a variety of configurations |
| Falcor3D [53] | 7 | 233,280 | single scene illuminated and viewed in varying conditions |
| Isaac3D [53] | 9 | 737,280 | synthetic robot arm holding objects in a variety of configurations |

Table 4: Shapes3D sources.

| index | description | values |
|---|---|---|
| 0 | floor color | 10 |
| 1 | object color | 10 |
| 2 | camera orientation | 10 |
| 3 | object scale | 8 |
| 4 | object shape | 4 |
| 5 | wall color | 15 |

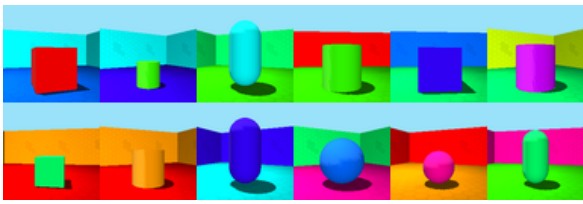

Figure 8: Shapes3D data samples.

Table 5: MPI3D sources.

| index | description | values |
|---|---|---|
| 0 | object color | 4 |
| 1 | object shape | 4 |
| 2 | object size | 2 |
| 3 | camera height | 3 |
| 4 | background color | 3 |
| 5 | robot x | 40 |
| 6 | robot y | 40 |

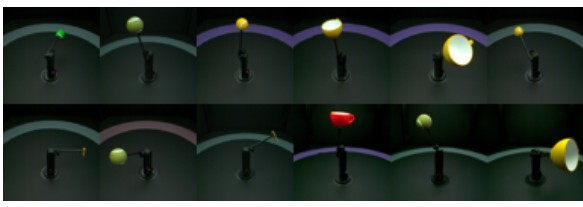

Figure 9: MPI3D data samples.

Table 6: Falcor3D sources.

| index | description | values |
|---|---|---|
| 0 | lighting intensity | 5 |
| 1 | lighting x | 6 |
| 2 | lighting y | 6 |
| 3 | lighting z | 6 |
| 4 | camera x | 6 |
| 5 | camera y | 6 |
| 6 | camera z | 6 |

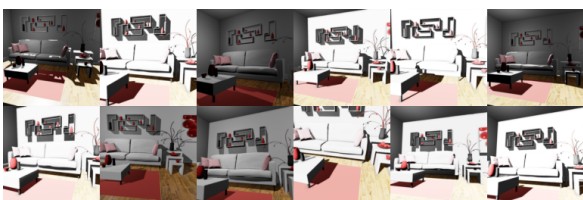

Figure 10: Falcor3D data samples.

Table 7: Isaac3D sources.

| index | description | values |
|---|---|---|
| 0 | object shape | 3 |
| 1 | robot x | 8 |
| 2 | robot y | 5 |
| 3 | camera height | 4 |
| 4 | object scale | 4 |
| 5 | lighting intensity | 4 |
| 6 | lighting direction | 6 |
| 7 | object color | 4 |
| 8 | wall color | 4 |

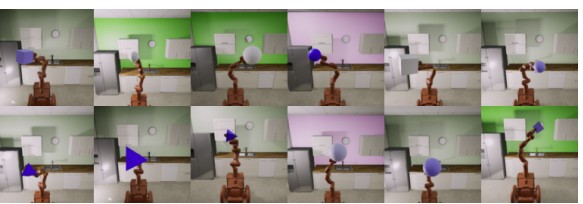

Figure 11: Isaac3D data samples.

## C.2 Hyperparameters

This section specifies fixed and tuned hyperparameters for all methods considered.

Table 8: Fixed latent quantization hyperparameters for QLAE and QLInfoWGAN-GP.

| hyperparameter | value |
|---|---|
| $n_v$ | 10 |
| $V_j$ initialization | $\text{linspace}(-0.5, 0.5, n_v)$ |
| $\lambda_{\text{quantize}}$ | $1 \times 10^{-2}$ |
| $\lambda_{\text{commit}}$ | $1 \times 10^{-2}$ |

Table 9: Fixed hyperparameters for all autoencoder variants.

| hyperparameter | value |
|---|---|
| $n_z$ | $2n_s$ |
| AdamW learning rate | $1 \times 10^{-3}$ |
| AdamW $\beta_1$ | 0.9 |
| AdamW $\beta_2$ | 0.99 |
| AdamW updates | $\leq 2 \times 10^5$ |
| batch size | 128 |
| $\lambda_{\text{reconstruct data}}$ | 1 |

Table 10: Fixed hyperparameters for all InfoGAN variants.

| hyperparameter | value |
|---|---|
| $n_z$ | $2n_s$ |
| AdamW learning rate | $2 \times 10^{-4}$ |
| AdamW $\beta_1$ | 0 |
| AdamW $\beta_2$ | 0.9 |
| AdamW updates | $\leq 2 \times 10^5$ |
| batch size | 64 |
| $\lambda_{\text{reconstruct latent}}$ | 200 |
| $\lambda_{\text{gradient penalty}}$ | 10 |
| $\lambda_{\text{value}}$ | 1 |

Table 11: Key regularization hyperparameter tuning done for each autoencoder and InfoGAN variant.

| method | hyperparameter | values |
|---|---|---|
| AE | weight decay | $[0, 0.001, 0.01, 0.1]$ |
| BioAE [72] | $\lambda_{\text{activity}}$ | $[0.01, 0.1, 1, 10, 100]$ |
| $\beta$-VAE [25] | $\beta = \lambda_{\text{KL}}$ | $[0.1, 0.3, 1, 3, 10]$ |
| $\beta$-TCVAE [11] | $\beta = \lambda_{\text{total correlation}}$ | $[0.1, 0.3, 1, 3, 10]$ |
| VQ-VAE [55] | weight decay | $[0.001, 0.01, 0.1, 1]$ |
| QLAE (ours) | weight decay | $[0.001, 0.01, 0.1, 1]$ |
| InfoWGAN-GP [54] | weight decay | $[0.0001, 0.001, 0.01, 0.1]$ |
| QLInfoWGAN-GP (ours) | weight decay | $[0.0001, 0.001, 0.01, 0.1]$ |

## C.3 Network Architectures

Inspired by recent works showing how well-designed decoder architectures can facilitate inductive biases relevant for disentanglement [33, 43], we use an expressive architecture for all results presented in this work.

**Encoder.** We use a simple feedforward convolutional encoder network. Each convolutional block consists of two resolution-preserving convolutional layers (kernel size 3, stride 1) and one downsampling convolutional layer (kernel size 4, stride 2) at a consistent width (number of channels). Each convolution operation is followed by a leaky ReLU (slope 0.3), then instance normalization. There are four such blocks with widths 32, 64, 128, and 256. The $256 \times 4 \times 4$ output is then flattened. Two dense layers each of width 256 with ReLU activation (and no normalization) follow. The final operation is an affine projection to the latent layer.

**Decoder.** We use a decoder architecture based on StyleGAN [33]. The latent code $z$ of shape $n_z$ passes through two dense layers each of width 256 with ReLU activation (and no normalization); call the output of this $w$. We directly parameterize a starting input feature map of shape $256 \times 4 \times 4$ with all entries initialized to $0.1$. Each style-decoding layer consists of processing an input feature map via a transposed convolution followed by a leaky ReLU (slope 0.3) and then an adaptive instance normalization (AdaIN): $w$ undergoes an affine projection to a scale and bias scalar for each channel of the output feature map, and the output feature map is instance normalized then affinely transformed by spatially broadcasting the scales and biases. Similar to the encoder, style-decoding layers are grouped into blocks, with each block consisting of two resolution-preserving style-decoding layers (kernel size 3, stride 1) and one upsampling style-decoding layer (kernel size 4, stride 2) at a consistent width. There are four such blocks with widths 256, 128, 64, and 32. The final operation is a $1 \times 1$ convolution of width 3 to yield an output of shape $3 \times 64 \times 64$.

## C.4 Negative Results

We tried a number of additional modifications to our basic methods, QLAE and QLInfoWGAN-GP, beyond the ablations presented in the main paper. Here is a list of those that only marginally helped, didn't help, or made things worse:

- Keeping the original VQ-VAE [55] hyperparameter settings for $\lambda_{\text{quantize}}$ (1) and $\lambda_{\text{commit}}$ (0.25).
- A step function schedule for the weight decay in AdamW: 0 for the first half of optimization, and a specified value for the second half.
- Linearly annealing $\lambda_{\text{quantize}}$ and/or $\lambda_{\text{commit}}$ from 0 up to a specified value.
- Using $\ell_1$ norm parameter vector regularization, in addition to or without weight decay.
- (QLInfoWGAN-GP-specific) Updating V with the encoder instead of with the decoder.

## C.5 Nonlinear DCI Evaluation

For each $p(\mathbf{s}_i \mid \mathbf{z})$ learning problem, we train a random forest classifier with 100 trees and the information gain splitting criterion. We use a 0.9/0.1 train/test split of the evaluation sample. We tune the maximum tree depth hyperparameter on held-out accuracy (see Figure 7 for an example of why this is important). We compute D and C using the most predictive model's relative feature importances, following their definitions [17]. We use held-out accuracy for I.

## D    Qualitative Results

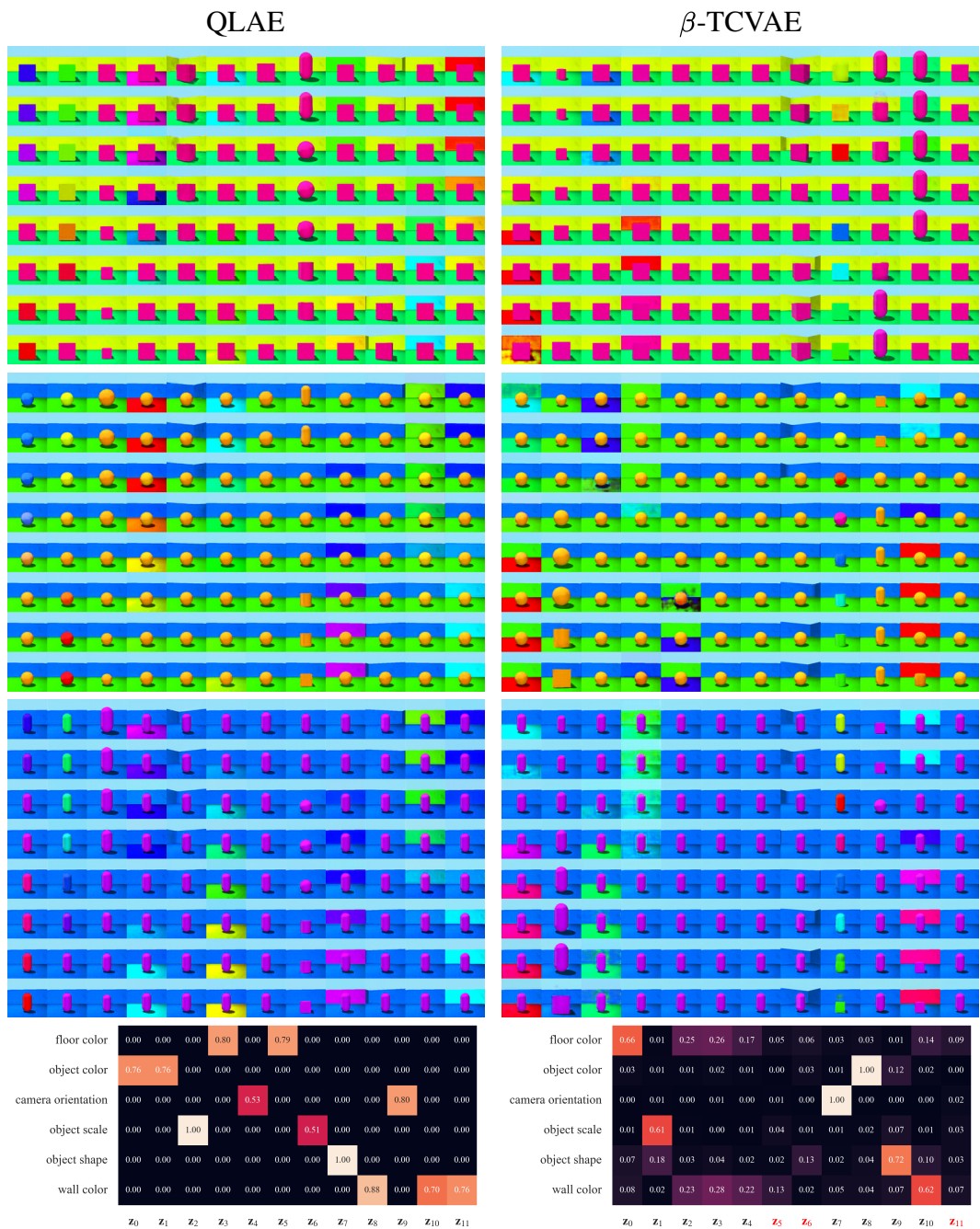

Figure 12: Decoded Shapes3D latent interventions and NMI heatmaps for QLAE and the prior method with highest InfoM. In each image block, a single data sample is encoded into its latent representation. In the $j$-th column, the $j$-th latent is intervened on with a linear interpolation of its range across the dataset. The resulting representation is then decoded. NMI heatmaps are annotated with source names, and inactive latents (as determined by range) have red font. In active latents, the visual variation in the generations caused by each latent tightly corresponds to sources that have significant NMI values in that latent's column. For example, the $\beta$-TCVAE's $z_1$ has high NMI with object scale and a lower NMI with object shape, and going down $z_1$'s column in the generations, we see consistent changes in object scale and occasional changes in object shape. Inactive latents, e.g. $\beta$-TCVAE's $z_5$, correspond to no discernable change in a column.

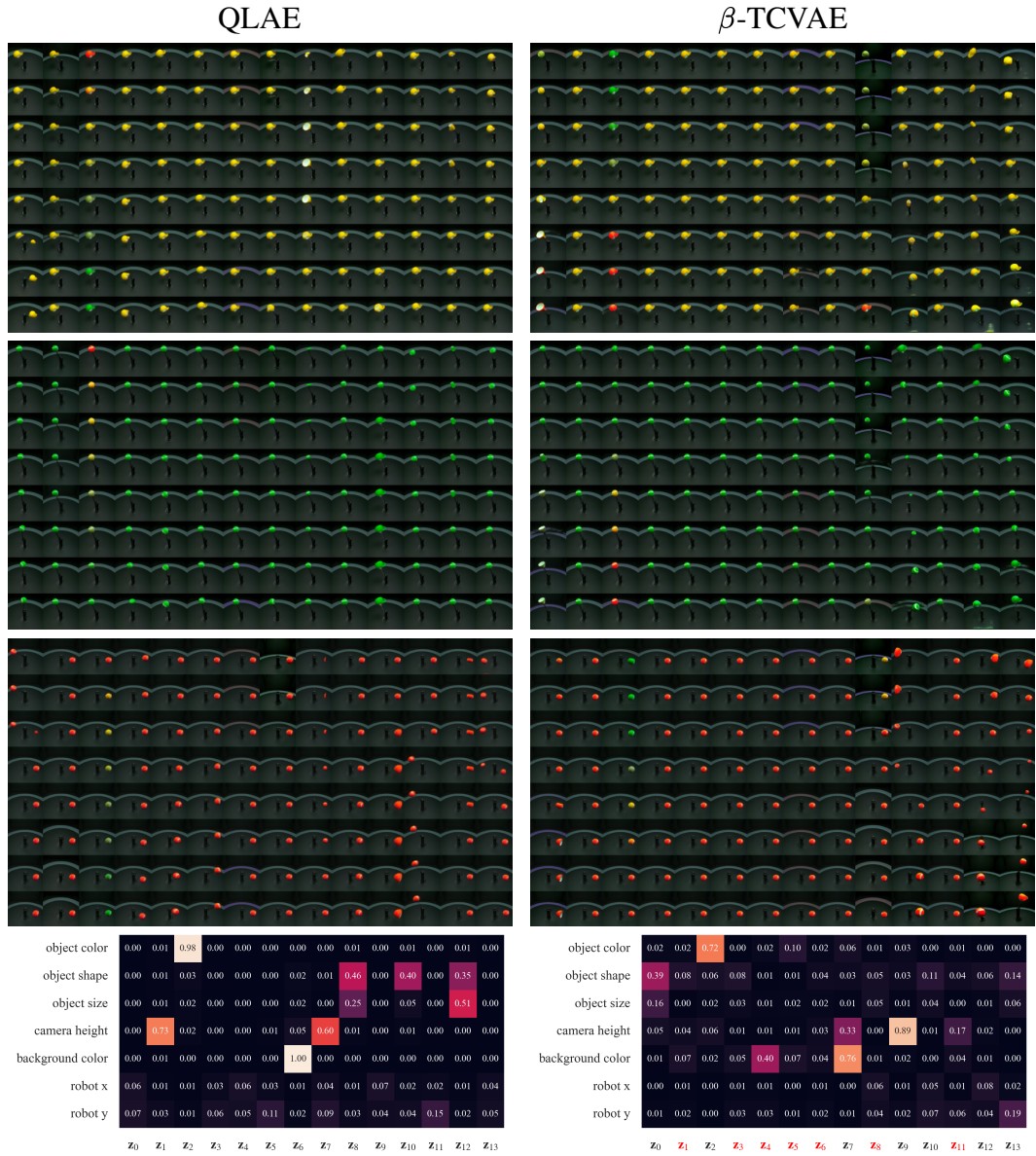

Figure 13: Decoded MPI3D latent interventions and NMI heatmaps for QLAE and the prior method with highest InfoM. In each image block, a single data sample is encoded into its latent representation. In the $j$-th column, the $j$-th latent is intervened on with a linear interpolation of its range across the dataset. The resulting representation is then decoded. NMI heatmaps are annotated with source names, and inactive latents (as determined by range) have red font. In active latents, the visual variation in the generations caused by each latent tightly corresponds to sources that have significant NMI values in that latent's column. For example, either QLAE's $\mathbf{z}_1$ (first and second data samples) or $\mathbf{z}_7$ (third data sample) causes changes in the camera height. The sensitivity of the NMI estimation can be observed in the low but non-negligible NMI values between the robot x and robot y sources and the QLAE's $\{\mathbf{z}_0, \mathbf{z}_3, \mathbf{z}_4, \mathbf{z}_5, \mathbf{z}_7, \mathbf{z}_9, \mathbf{z}_{11}, \mathbf{z}_{13}\}$ manifesting as rare or small changes in pose when intervening on those latents. This sensitivity, however, means that it is important to remove inactive latents from InfoMEC estimation: the $\beta$-TCVAE's $\mathbf{z}_4$ and $\mathbf{z}_{11}$ are estimated to have substantial NMI with background color and camera height, but do not actually cause any changes in the generations.

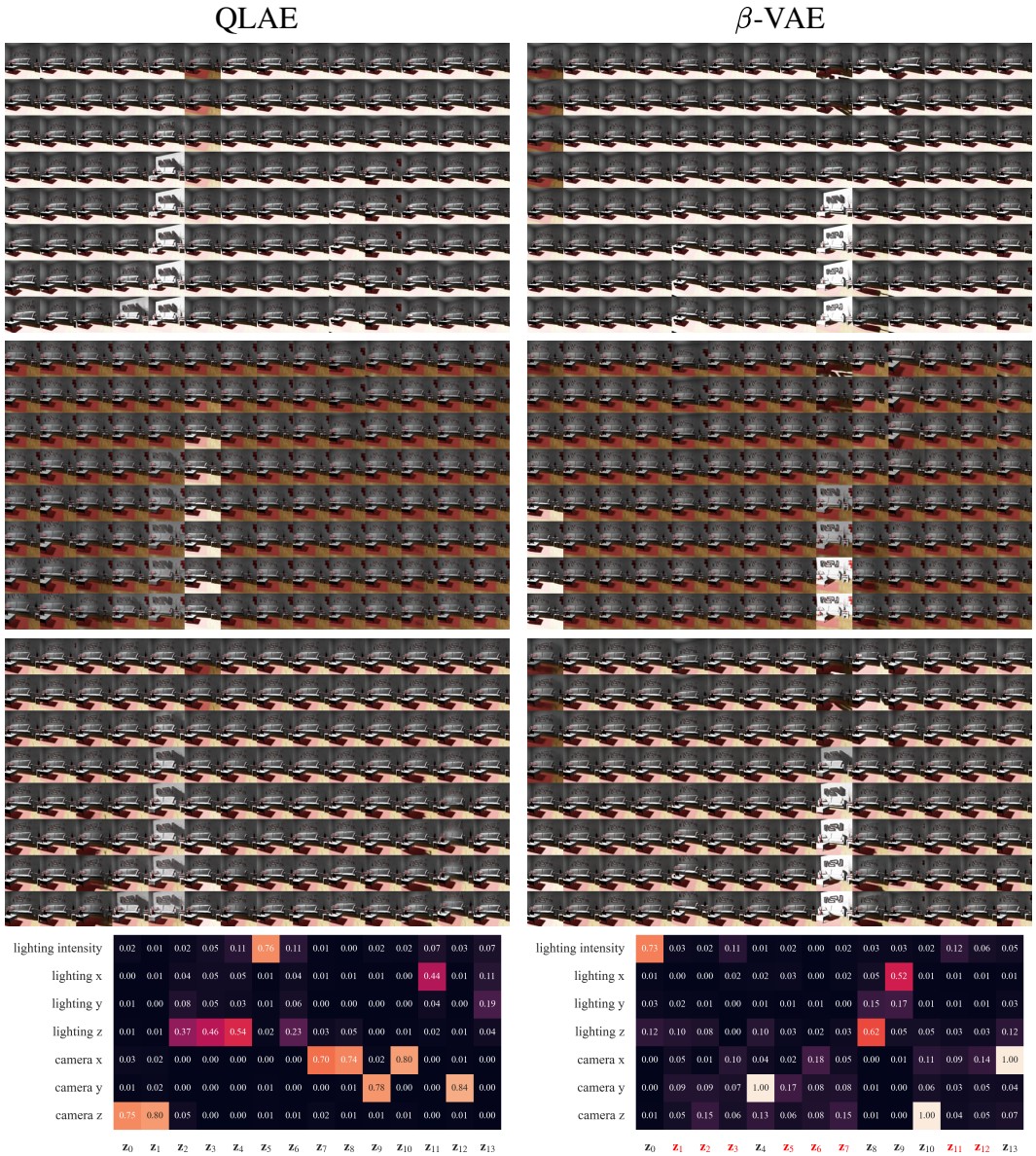

Figure 14: Decoded Falcor3D latent interventions and NMI heatmaps for QLAE and the prior method with highest InfoM. In each image block, a single data sample is encoded into its latent representation. In the $j$-th column, the $j$-th latent is intervened on with a linear interpolation of its range across the dataset. The resulting representation is then decoded. NMI heatmaps are annotated with source names, and inactive latents (as determined by range) have red font. In active latents, the visual variation in the generations caused by each latent tightly corresponds to sources that have significant NMI values in that latent's column. This holds for both obvious visual changes in sources like lighting z, and rather subtle visual changes in sources like camera x. Pruning inactive latents from NMI before computing InfoM and InfoC is particularly important for models that optimize for latent shrinkage, such as the $\beta$-VAE: (InfoM, InfoC) is $(0.71, 0.70)$ with pruning vs. $(0.47, 0.51)$ without. Pruning is justified by the decoded interventions showing no dependence on the inactive latents.

QLAE                                                    BioAE

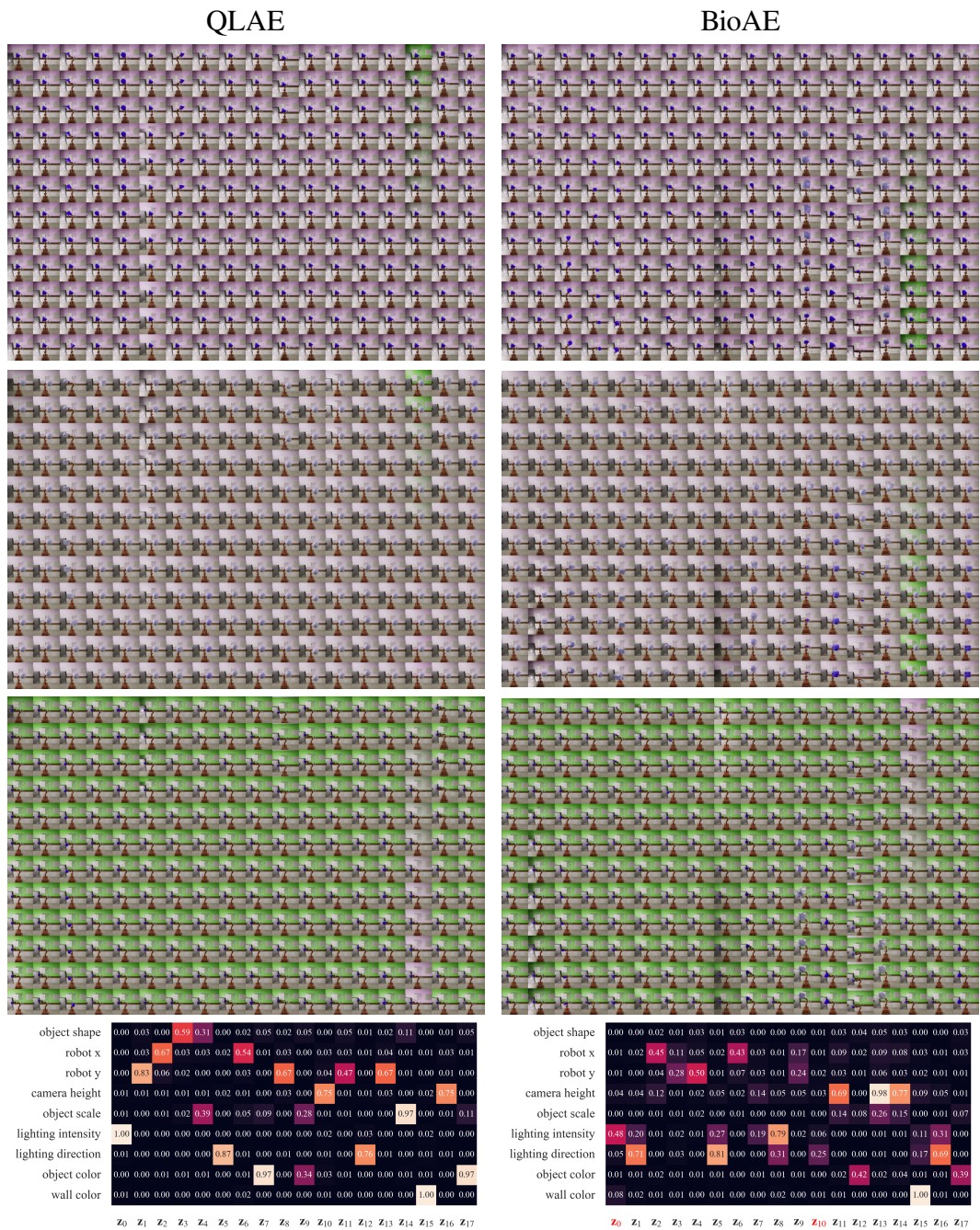

Figure 15: Decoded Isaac3D latent interventions and NMI heatmaps for QLAE and the prior method with highest InfoM. In each image block, a single data sample is encoded into its latent representation. In the $j$-th column, the $j$-th latent is intervened on with a linear interpolation of its range across the dataset. The resulting representation is then decoded. NMI heatmaps are annotated with source names, and inactive latents (as determined by range) have red font. In active latents, the visual variation in the generations caused by each latent tightly corresponds to sources that have significant NMI values in that latent's column.

# E    Quantitative Results

This section contains unabridged results for the experiments in the main text. Intervals denote 95% confidence intervals of the mean estimated assuming a $t$-distribution. Bolded intervals overlap with the interval with highest endpoint in the column. AE and InfoGAN variants are presented and bolded separately as all AEs are filtered for near-perfect data reconstruction, whereas InfoGANs are generally more lossy.

Table 12: Full results for the Shapes3D dataset.

| model | InfoM ↑ | InfoE ↑ | InfoC ↑ | D ↑ | I ↑ | C ↑ | PSNR (dB) ↑ |
|---|---|---|---|---|---|---|---|
| AE | $0.41 \pm 0.03$ | $\mathbf{0.98 \pm 0.01}$ | $0.28 \pm 0.01$ | $0.11 \pm 0.02$ | $0.82 \pm 0.01$ | $0.08 \pm 0.02$ | $\mathbf{37.3 \pm 0.1}$ |
| $\beta$-VAE | $0.59 \pm 0.02$ | $\mathbf{0.99 \pm 0.02}$ | $0.49 \pm 0.03$ | $0.61 \pm 0.02$ | $\mathbf{0.99 \pm 0.01}$ | $0.47 \pm 0.01$ | $35.1 \pm 0.1$ |
| $\beta$-TCVAE | $0.61 \pm 0.03$ | $0.82 \pm 0.02$ | $\mathbf{0.62 \pm 0.02}$ | $0.46 \pm 0.02$ | $\mathbf{0.99 \pm 0.01}$ | $0.38 \pm 0.01$ | $34.6 \pm 0.1$ |
| BioAE | $0.56 \pm 0.02$ | $\mathbf{0.98 \pm 0.01}$ | $0.44 \pm 0.02$ | $0.33 \pm 0.01$ | $0.93 \pm 0.01$ | $0.25 \pm 0.01$ | $31.6 \pm 0.1$ |
| VQ-VAE | $0.72 \pm 0.03$ | $\mathbf{0.97 \pm 0.02}$ | $0.47 \pm 0.03$ | $0.40 \pm 0.01$ | $0.84 \pm 0.02$ | $0.34 \pm 0.01$ | $36.0 \pm 0.0$ |
| QLAE (ours) | $\mathbf{0.95 \pm 0.02}$ | $\mathbf{0.99 \pm 0.01}$ | $0.55 \pm 0.02$ | $\mathbf{0.81 \pm 0.01}$ | $\mathbf{0.99 \pm 0.01}$ | $\mathbf{0.61 \pm 0.01}$ | $32.9 \pm 0.2$ |
| QLAE w/ global codebook | $\mathbf{0.96 \pm 0.01}$ | $\mathbf{0.99 \pm 0.00}$ | $0.48 \pm 0.01$ | $\mathbf{0.83 \pm 0.02}$ | $\mathbf{0.99 \pm 0.01}$ | $\mathbf{0.60 \pm 0.01}$ | $34.8 \pm 0.2$ |
| QLAE w/o weight decay | $0.69 \pm 0.02$ | $\mathbf{0.99 \pm 0.01}$ | $0.51 \pm 0.02$ | $0.63 \pm 0.02$ | $\mathbf{0.99 \pm 0.01}$ | $0.47 \pm 0.03$ | $\mathbf{37.5 \pm 0.3}$ |
| VQ-VAE w/ weight decay | $0.80 \pm 0.01$ | $\mathbf{0.99 \pm 0.01}$ | $0.46 \pm 0.02$ | $0.74 \pm 0.01$ | $\mathbf{0.99 \pm 0.01}$ | $0.57 \pm 0.01$ | $\mathbf{37.2 \pm 0.1}$ |
| InfoWGAN-GP | $0.61 \pm 0.02$ | $\mathbf{0.78 \pm 0.03}$ | $0.41 \pm 0.02$ | $0.23 \pm 0.01$ | $0.80 \pm 0.01$ | $0.18 \pm 0.01$ | $12.0 \pm 0.6$ |
| QLInfoWGAN-GP (ours) | $\mathbf{0.73 \pm 0.03}$ | $\mathbf{0.75 \pm 0.03}$ | $\mathbf{0.48 \pm 0.02}$ | $\mathbf{0.38 \pm 0.01}$ | $\mathbf{0.85 \pm 0.01}$ | $\mathbf{0.29 \pm 0.01}$ | $\mathbf{20.4 \pm 0.4}$ |
| QLInfoWGAN-GP w/o w.d. | $0.66 \pm 0.03$ | $0.57 \pm 0.05$ | $0.40 \pm 0.04$ | $0.16 \pm 0.01$ | $0.71 \pm 0.02$ | $0.13 \pm 0.02$ | $16.9 \pm 0.7$ |

Table 13: Full results for the MPI3D dataset.

| model | InfoM ↑ | InfoE ↑ | InfoC ↑ | D ↑ | I ↑ | C ↑ | PSNR (dB) ↑ |
|---|---|---|---|---|---|---|---|
| AE | $0.37 \pm 0.04$ | $0.72 \pm 0.03$ | $0.36 \pm 0.03$ | $0.15 \pm 0.02$ | $\mathbf{0.83 \pm 0.02}$ | $0.14 \pm 0.02$ | $36.2 \pm 0.2$ |
| $\beta$-VAE | $0.45 \pm 0.03$ | $0.71 \pm 0.03$ | $\mathbf{0.51 \pm 0.03}$ | $\mathbf{0.31 \pm 0.02}$ | $\mathbf{0.83 \pm 0.02}$ | $0.27 \pm 0.02$ | $\mathbf{38.4 \pm 0.2}$ |
| $\beta$-TCVAE | $0.51 \pm 0.04$ | $0.60 \pm 0.04$ | $\mathbf{0.57 \pm 0.04}$ | $0.22 \pm 0.02$ | $\mathbf{0.77 \pm 0.02}$ | $0.21 \pm 0.02$ | $37.1 \pm 0.2$ |
| BioAE | $0.45 \pm 0.03$ | $0.66 \pm 0.04$ | $0.36 \pm 0.03$ | $0.24 \pm 0.02$ | $\mathbf{0.79 \pm 0.02}$ | $0.19 \pm 0.02$ | $37.9 \pm 0.2$ |
| VQ-VAE | $0.43 \pm 0.06$ | $0.57 \pm 0.04$ | $0.22 \pm 0.04$ | $0.09 \pm 0.02$ | $0.63 \pm 0.02$ | $0.14 \pm 0.03$ | $35.3 \pm 0.2$ |
| QLAE (ours) | $\mathbf{0.61 \pm 0.04}$ | $0.63 \pm 0.05$ | $\mathbf{0.51 \pm 0.03}$ | $\mathbf{0.36 \pm 0.04}$ | $\mathbf{0.85 \pm 0.04}$ | $\mathbf{0.36 \pm 0.05}$ | $37.2 \pm 0.6$ |
| QLAE w/ global codebook | $\mathbf{0.54 \pm 0.04}$ | $0.62 \pm 0.02$ | $0.45 \pm 0.03$ | $\mathbf{0.36 \pm 0.04}$ | $\mathbf{0.86 \pm 0.02}$ | $\mathbf{0.34 \pm 0.02}$ | $37.2 \pm 0.6$ |
| QLAE w/o weight decay | $0.51 \pm 0.03$ | $0.61 \pm 0.02$ | $0.48 \pm 0.04$ | $\mathbf{0.30 \pm 0.04}$ | $\mathbf{0.84 \pm 0.06}$ | $\mathbf{0.29 \pm 0.03}$ | $37.7 \pm 0.3$ |
| VQ-VAE w/ weight decay | $0.50 \pm 0.04$ | $\mathbf{0.81 \pm 0.04}$ | $0.41 \pm 0.04$ | $0.22 \pm 0.02$ | $0.68 \pm 0.02$ | $0.20 \pm 0.01$ | $34.7 \pm 0.2$ |
| InfoWGAN-GP | $0.43 \pm 0.04$ | $0.40 \pm 0.05$ | $0.20 \pm 0.05$ | $0.09 \pm 0.03$ | $0.63 \pm 0.01$ | $0.09 \pm 0.02$ | $\mathbf{25.2 \pm 1.4}$ |
| QLInfoWGAN-GP (ours) | $\mathbf{0.62 \pm 0.03}$ | $\mathbf{0.51 \pm 0.04}$ | $\mathbf{0.37 \pm 0.04}$ | $\mathbf{0.24 \pm 0.02}$ | $\mathbf{0.71 \pm 0.02}$ | $\mathbf{0.25 \pm 0.02}$ | $\mathbf{26.3 \pm 1.5}$ |
| QLInfoWGAN-GP w/o w.d. | $\mathbf{0.63 \pm 0.04}$ | $\mathbf{0.55 \pm 0.04}$ | $\mathbf{0.40 \pm 0.05}$ | $\mathbf{0.28 \pm 0.01}$ | $\mathbf{0.74 \pm 0.03}$ | $\mathbf{0.23 \pm 0.03}$ | $\mathbf{26.3 \pm 1.5}$ |

Table 14: Full results for the Falcor3D dataset.

| model | InfoM ↑ | InfoE ↑ | InfoC ↑ | D ↑ | I ↑ | C ↑ | PSNR (dB) ↑ |
|---|---|---|---|---|---|---|---|
| AE | $0.39 \pm 0.03$ | $0.74 \pm 0.03$ | $0.20 \pm 0.03$ | $0.08 \pm 0.02$ | $0.76 \pm 0.02$ | $0.07 \pm 0.01$ | $\mathbf{29.9 \pm 0.1}$ |
| $\beta$-VAE | $\mathbf{0.71 \pm 0.05}$ | $0.73 \pm 0.04$ | $\mathbf{0.70 \pm 0.03}$ | $0.32 \pm 0.02$ | $0.84 \pm 0.02$ | $0.28 \pm 0.01$ | $29.0 \pm 0.2$ |
| $\beta$-TCVAE | $0.66 \pm 0.02$ | $0.74 \pm 0.04$ | $\mathbf{0.71 \pm 0.04}$ | $0.36 \pm 0.02$ | $0.90 \pm 0.02$ | $0.33 \pm 0.01$ | $\mathbf{29.6 \pm 0.1}$ |
| BioAE | $0.54 \pm 0.05$ | $0.73 \pm 0.04$ | $0.31 \pm 0.01$ | $0.21 \pm 0.02$ | $0.81 \pm 0.02$ | $0.17 \pm 0.02$ | $29.4 \pm 0.1$ |
| VQ-VAE | $0.61 \pm 0.04$ | $\mathbf{0.83 \pm 0.05}$ | $0.42 \pm 0.02$ | $0.30 \pm 0.03$ | $0.79 \pm 0.01$ | $0.29 \pm 0.01$ | $29.0 \pm 0.1$ |
| QLAE (ours) | $\mathbf{0.71 \pm 0.03}$ | $0.77 \pm 0.02$ | $0.44 \pm 0.02$ | $\mathbf{0.50 \pm 0.03}$ | $\mathbf{0.96 \pm 0.03}$ | $\mathbf{0.38 \pm 0.02}$ | $29.4 \pm 0.4$ |
| QLAE w/ global codebook | $0.59 \pm 0.01$ | $0.74 \pm 0.03$ | $0.37 \pm 0.01$ | $0.36 \pm 0.02$ | $0.90 \pm 0.02$ | $0.26 \pm 0.04$ | $28.4 \pm 0.4$ |
| QLAE w/o weight decay | $0.65 \pm 0.02$ | $0.76 \pm 0.02$ | $0.43 \pm 0.02$ | $\mathbf{0.46 \pm 0.02}$ | $\mathbf{0.97 \pm 0.02}$ | $\mathbf{0.36 \pm 0.02}$ | $\mathbf{29.6 \pm 0.4}$ |
| VQ-VAE w/ weight decay | $\mathbf{0.74 \pm 0.02}$ | $\mathbf{0.86 \pm 0.04}$ | $0.40 \pm 0.03$ | $0.41 \pm 0.02$ | $0.85 \pm 0.01$ | $0.32 \pm 0.02$ | $\mathbf{29.1 \pm 0.1}$ |
| InfoWGAN-GP | $0.44 \pm 0.04$ | $\mathbf{0.60 \pm 0.03}$ | $0.30 \pm 0.04$ | $0.11 \pm 0.01$ | $\mathbf{0.74 \pm 0.01}$ | $0.08 \pm 0.02$ | $18.9 \pm 0.6$ |
| QLInfoWGAN-GP (ours) | $\mathbf{0.54 \pm 0.04}$ | $0.53 \pm 0.04$ | $\mathbf{0.56 \pm 0.03}$ | $\mathbf{0.20 \pm 0.01}$ | $0.73 \pm 0.02$ | $\mathbf{0.24 \pm 0.02}$ | $\mathbf{17.6 \pm 1.0}$ |
| QLInfoWGAN-GP w/o w.d. | $\mathbf{0.56 \pm 0.03}$ | $0.56 \pm 0.04$ | $\mathbf{0.52 \pm 0.02}$ | $0.14 \pm 0.02$ | $0.72 \pm 0.02$ | $0.17 \pm 0.02$ | $\mathbf{17.6 \pm 1.1}$ |

Table 15: Full results for the Isaac3D dataset.

| model | InfoM ↑ | InfoE ↑ | InfoC ↑ | D ↑ | I ↑ | C ↑ | PSNR (dB) ↑ |
|---|---|---|---|---|---|---|---|
| AE | $0.42 \pm 0.04$ | $0.80 \pm 0.02$ | $0.21 \pm 0.05$ | $0.13 \pm 0.02$ | $0.85 \pm 0.02$ | $0.10 \pm 0.02$ | $38.4 \pm 0.1$ |
| $\beta$-VAE | $0.60 \pm 0.03$ | $0.80 \pm 0.02$ | $\mathbf{0.51 \pm 0.03}$ | $0.23 \pm 0.02$ | $0.88 \pm 0.02$ | $0.19 \pm 0.01$ | $39.8 \pm 0.1$ |
| $\beta$-TCVAE | $0.54 \pm 0.02$ | $0.70 \pm 0.02$ | $\mathbf{0.46 \pm 0.03}$ | $0.19 \pm 0.02$ | $0.84 \pm 0.02$ | $0.16 \pm 0.01$ | $38.2 \pm 0.1$ |
| BioAE | $0.63 \pm 0.03$ | $0.65 \pm 0.03$ | $0.33 \pm 0.04$ | $0.38 \pm 0.02$ | $0.91 \pm 0.01$ | $0.31 \pm 0.02$ | $38.0 \pm 0.2$ |
| VQ-VAE | $0.57 \pm 0.04$ | $0.87 \pm 0.05$ | $\mathbf{0.45 \pm 0.04}$ | $0.33 \pm 0.02$ | $0.89 \pm 0.02$ | $0.31 \pm 0.01$ | $39.8 \pm 0.1$ |
| QLAE (ours) | $\mathbf{0.78 \pm 0.03}$ | $\mathbf{0.97 \pm 0.03}$ | $0.49 \pm 0.03$ | $\mathbf{0.69 \pm 0.02}$ | $\mathbf{0.99 \pm 0.01}$ | $\mathbf{0.54 \pm 0.02}$ | $\mathbf{40.4 \pm 0.3}$ |
| QLAE w/ global codebook | $0.65 \pm 0.03$ | $0.82 \pm 0.02$ | $\mathbf{0.46 \pm 0.03}$ | $0.53 \pm 0.03$ | $\mathbf{0.99 \pm 0.02}$ | $0.43 \pm 0.03$ | $39.1 \pm 0.4$ |
| QLAE w/o weight decay | $0.66 \pm 0.02$ | $0.89 \pm 0.03$ | $\mathbf{0.51 \pm 0.02}$ | $0.58 \pm 0.04$ | $\mathbf{0.99 \pm 0.01}$ | $0.48 \pm 0.04$ | $\mathbf{40.9 \pm 0.3}$ |
| VQ-VAE w/ weight decay | $\mathbf{0.73 \pm 0.03}$ | $0.81 \pm 0.03$ | $0.44 \pm 0.04$ | $0.34 \pm 0.02$ | $0.85 \pm 0.02$ | $0.39 \pm 0.01$ | $35.3 \pm 0.1$ |
| InfoWGAN-GP | $0.53 \pm 0.03$ | $0.51 \pm 0.02$ | $0.24 \pm 0.02$ | $0.13 \pm 0.02$ | $0.71 \pm 0.02$ | $0.11 \pm 0.02$ | $21.9 \pm 0.9$ |
| QLInfoWGAN-GP (ours) | $\mathbf{0.63 \pm 0.03}$ | $\mathbf{0.58 \pm 0.03}$ | $\mathbf{0.49 \pm 0.03}$ | $\mathbf{0.24 \pm 0.01}$ | $\mathbf{0.79 \pm 0.01}$ | $\mathbf{0.25 \pm 0.01}$ | $\mathbf{25.3 \pm 0.7}$ |
| QLInfoWGAN-GP w/o w.d. | $\mathbf{0.59 \pm 0.01}$ | $\mathbf{0.53 \pm 0.04}$ | $\mathbf{0.45 \pm 0.04}$ | $\mathbf{0.20 \pm 0.03}$ | $\mathbf{0.77 \pm 0.02}$ | $\mathbf{0.23 \pm 0.01}$ | $23.2 \pm 1.1$ |

