# OpenReview forum: "Disentanglement via Latent Quantization"
_NeurIPS.cc/2023/Conference — NeurIPS 2023 poster_

### Official Review · Reviewer_Mf8i · 2023-07-03

**Soundness:** 2 fair
**Presentation:** 2 fair
**Contribution:** 2 fair
**Rating:** 5
**Confidence:** 2

**Summary:**

The paper proposes learning a disentangled latent representation by quantizing the latent space into learnable discrete values, with a separate scalar codebook for each dimension. The main assumption that the paper relies on is that "many datasets of interest are generated from their sources in a compositional manner." Additionally, the paper introduces a new metric to better evaluate the disentanglement.

**Strengths:**

The paper proposed a new metric for disentanglement.


**Weaknesses:**



1) The manuscript could benefit from improvements as some parts and notations are rather confusing.

2) The proposed method appears to be very similar to VQ-VAE.

3) It is uncertain if the underlying assumption that real-world data is generated in a compositional manner holds.

4) Concerning disentanglement, the paper only presents results on the metric they propose. It would be interesting to see a comparison with VQ-VAE using other disentanglement metrics.

**Questions:**

1) In line 49, the author mentions that "they enable the use of simpler, more robust, distribution estimation techniques than continuous latent spaces." Could you elaborate on that?

2) The new interpretation given in equation 3 for infoGAN does not appear convincing.

3) In line 121, it is mentioned that "generative models for realistic data are compositional." What is meant by generative models being compositional? Earlier, it was mentioned that realistic data is generated in a compositional manner. I assume these two statements imply the same thing. I wonder if this assumption oversimplifies the nature of generative models and may not be generalizable.

**Limitations:**

The limitation of the paper is discussed and there is no potential negative societal impact of their work.

---

> ### Author Rebuttal · Authors · 2023-08-10
>
> Thank you for your review. We believe we can address the main concerns you had about our submission, in particular the similarity to VQ-VAE and only evaluating with InfoMEC.
>
> > The proposed method appears to be very similar to VQ-VAE.
>
> Indeed, latent quantization can be summarized as three modifications to vector quantization: i) scalar instead of vector codes, ii) atypically high weight decay on the encoder and decoder, and iii) separate per-dimension codebooks instead of a single global codebook. Each of these changes is ablated with our full experimental protocol on autoencoders and all are shown to be necessary for best performance (Table 5). Even if one views these contributions as not constituting much algorithmic novelty, we feel there is substantial impact in identifying and verifying a recipe for dramatic improvement on a difficult problem. In any case, we take care to make proper attributions to the VQ-VAE work in the technical section, algorithm pseudocode, and related works section. We have also refined our Figure 2 to more clearly delineate the differences between vector quantization and latent quantization.
>
> > It is uncertain if the underlying assumption that real-world data is generated in a compositional manner holds.
>
> While there may exist datasets that aren’t accurately described by this assumption, it certainly holds for the ones we (and the field of disentangled representation learning) consider. For example, take the “complex shapes” variant of MPI3D we use, which consists of images taken of a real world robotics apparatus. Changing any particular source dimension while holding the others fixed corresponds to a consistent and modular change in the data, e.g. via physically actuating the manipulator, swapping out the object, or moving the camera.
>
> > Concerning disentanglement, the paper only presents results on the metric they propose. It would be interesting to see a comparison with VQ-VAE using other disentanglement metrics.
>
> Thank you for pointing this out. We have now added random forest DCI evaluation. Please see the "Quantitative results in terms of other disentanglement metrics" section of our main rebuttal, as well as the rebuttal pdf for tables.
>
> In particular, the improvement of QLAE over VQ-VAE holds under DCI, too.
>
> > In line 49, the author mentions that "they enable the use of simpler, more robust, distribution estimation techniques than continuous latent spaces." Could you elaborate on that?
>
> When a source and latent are both discrete, their joint distribution can be characterized by explicitly constructing a table of frequencies over an evaluation dataset. This is maximally expressive, so we are freed from needing to make any approximating modeling choices, such as binning strategy, number of nearest neighbors, kernel, or variational family. We briefly mention this advantage in the context of mutual information estimation in Section 4.1 (specifically line 140).
>
> > The new interpretation given in equation 3 for infoGAN does not appear convincing.
>
> The interpretation of the InfoGAN encoder loss as latent reconstruction serves to justify practical implementations (e.g. mean-squared error) as natural design choices rather than loose variational bounds on mutual information. The interpretation also provides a lens with which to see autoencoders and InfoGANs as duals, motivating our demonstration of the applicability of latent quantization to both in our experiments. Since these points are entirely tangential to the submission’s main contributions, we opted for a very condensed statement; our apologies if this detracted from the reading experience. Note also that the interpretation aligns closely with the method pseudocode (Algorithm 3, Appendix A).
>
> > “generative models for realistic data are compositional”
>
> Thank you for pointing this out. There is a typo of “generative models”, which we have amended to “generative processes”, i.e. the real world (or high-fidelity simulation) physics that generate the data. We hope this clears up the confusion.
>
> > The manuscript could benefit from improvements as some parts and notations are rather confusing.
>
> If there is anything specific beyond what you already listed as questions, please do let us know, as your constructive feedback is much appreciated.

---

> > ### Comment · Reviewer_Mf8i · 2023-08-18
> > **Rebuttal Acknowledgment**
> >
> > I appreciate the author's comprehensive response to my inquiries. While many of my concerns have been addressed, having gone through all the reviews and comments, I still have my concerns over the contribution compared to the VQVAE. As a result, I have chosen to maintain my current score.

---

> > > ### Author Response · Authors · 2023-08-21
> > >
> > > Thank you for informing us that we have addressed many of your concerns, and for explaining your decision-making. We find your remaining reservation sensible and imagine that others may share it, so we feel it prudent to expand on the relationship between VQ-VAE and our work.
> > >
> > > The original VQ-VAE work’s experiments focused on generative modeling. Our work adapts the discrete representation learning techniques introduced for VQ-VAE to the related but separate problem of unsupervised disentanglement, which goes beyond generative modeling in asking for the learned representations to exhibit a specific *functional* relationship to the data and its underlying generative sources. This distinction is far from trivial: it manifests in a markedly different evaluation protocol in terms of datasets, metrics, and even desired qualitative properties. **Part of our contribution, then, is in the novel use of quantized representations (à la VQ-VAE) for better disentanglement.** This is noted by Reviewers sKhi and XVcp in their original reviews.
> > >
> > > Why hasn’t this been shown previously, given all the attention and adoption that VQ-VAE has deservedly enjoyed? One potential reason can be seen in Tables 1 and 2 of the rebuttal pdf: **Out-of-the-box VQ-VAE simply doesn’t disentangle very well**. Its modularity and explicitness is only somewhat competitive with prior methods under both DCI and InfoMEC. In contrast, QLAE dominates in modularity (D and InfoM), the most important disentanglement property. The three modifications to VQ-VAE that enable this result are indeed embarrassingly simple, but, in deep learning, such details matter, and our ablation studies verify this. The fact that QLAE is the amalgamation of existing, popular techniques does limit the technical novelty in this work, but it also increases the potential impact via easing adoption and providing connections to previous work.
> > >
> > > For completeness, we also remark on a few quality of life differences. VQ-VAE’s categorical latents cause additional bookkeeping during evaluation and leave any ordinal structure in each dimension unspecified. We tried using the high-dimensional discrete decoder input for evaluation, but this unsurprisingly resulted in poor performance across the board. In contrast, QLAE’s latent space can be treated identically to the continuous latents of prior methods, e.g. for DCI evaluation, or for visualizing interpolating interventions on latents (as shown in the rebuttal pdf).
> > >
> > > We would greatly appreciate your consideration of these points. Thank you again for the time and effort you have spent on reviewing this work!

---

### Official Review · Reviewer_XVcp · 2023-07-04

**Soundness:** 3 good
**Presentation:** 2 fair
**Contribution:** 2 fair
**Rating:** 7
**Confidence:** 5

**Summary:**

The paper presents an alternative approach to the standard disentanglement representations learning framework whereby the continuous latent space is substituted by a quantized one. Based on this new quantized latent space, the authors propose novel ways to measuring several properties related to disentanglement. They then make two main claims: 1) that this quantized latent space better aligns with how generative factors are usually encountered in data (both real and synthetic) and 2) that their novel disentangled measures more robustly measure the real disentanglement learned by the model.

**Strengths:**

In exploring the idea of using a quantised latent space to learn a disentangled representation of the data, the paper has several key strengths:
1. The authors use a good theoretical motivation based on results from the ICA literature.
2. The observation that disentangled datasets use quantized generative factors (even if the values are intended to be over the reals) is one that, even if obvious, not many researchers have explored (though some have, see below).
3. The disentanglement metrics are well motivated based on this quantized representation.
4. The authors used two main variants and several datasets to test their approach which gives their results robustness.

**Weaknesses:**

On the other hand there are several places where the paper as is currently written is lacking in my opinion. In no particular order these are:

1. While the authors present numerical results for five datasets, they only provide traversal reconstructions for one, visualization of one metric for one dataset and no visualization of the actual latent values.
2. Despite the fact that the proposed disentanglement measurement framework bears a significant resemblance to the one proposed in Eastwood and Williams, they only reference this prior work in passing without giving any in depth comparison between the two.
3. The authors make the point several times that they are using a compositional representation (e.g. with phrases such as “*A* *side advantage of the compositional latent encoding…”)* yet they seem to misunderstand what compositionality is and whether their experiments test for this.
4. There is no discussion of any limitations of their approach even though some of them are obvious (like the fact that we need to define the size of the quantization).

**Questions:**

For the first point, the authors claimed to have tested 5 datasets and two variants of their model against some previous models. However, the scores for their framework are only provided for the 3DShapes dataset. While it is true that space is limited, I would have expected them to include the other plots in the appendix. Similarly, I would have expected to see more traversal reconstructions for the other datasets in the appendix. Speaking of which the authors left a comment in big red bright letters which I assume was for their own benefit. Always review before submitting!

I assume including these plots in the appendix will be easy enough. What I do expect to see is a visualization of the latent space of the model (even if it is only for arbitrary pairs of variables). They clearly visualize how the latent space should look for their model and then… proceed to not show how it actually looks for trained models. This should have been one of their core plots. Yes, metrics are better for comparing a large number of models, but visual inspection can offer important insights and one of their core contributions is that a quantized latent space should be more transparent.

The second issue is that their InfoMEC framawork is very similar to the DCI framework of Eastwood & Williams (2019), whose concepts of disentanglement and completeness are equivalent to those compactness and modularity, respectively. A comparison between the two is thus warranted to prove the effectiveness of their approach. This not only requires overall scores but NMI visualizations and visualizations of the latent space in order to check that the authors claims (i.e. that their metrics are better) are indeed true.

**Limitations:**

There is also a lack of discussion on some of the limitations of this approach. One reason why a continuous space is often preferred is because it is not necessary to set any other parameters. While I agree with the authors hypothesis that a quantized space should be better, it also introduced the problem of setting the size of this quantized space. What if we do not know how many sources there are? The authors need to at least discuss how this is a limitation of their proposal.

Finally, there is a recurring conceptual issue in the introduction, namely the conflation of disentanglement (which is what they authors explore) with compositionality (which they are not). While it is certainly possible to take inspiration from how generative factors in disentanglement datasets are defined (and this is indeed what the authors do), the fact that those representations are compositional does not entail that the quantized space that the authors propose will also be compositional.

To see why notice that the definition of compositionality is that a system uses its knowledge of existing entities and rules of composition of a particular domain to understand novel instances (see [1]). Thus compositionality is a mechanism to transfer performance from training to out-of-distribution settings. All the experiments in the current work are in-distribution and thus do not test for compositionality.

In the case of the ground truth generative factors, these are compositional because they are used by very specific image generators that understand the semantics of these values in the context of the images they must generate. Whether the decoder used by VAEs has this capacity (which would make the representations compositional) is something that the authors do not test (and is actually not the case! See below).

I do not expect the authors to explore the compositionality issue here as this would be an entirely different article. But they should be able to partly rewrite the introduction accordingly, limiting their references to compositionality to the very minimal (we take inspiration from…) and refrain from referring to the quantized latent space as compositional.

If the authors are interested I would recommend Duan et al., for a hypothesis of the relations between disentanglement and compositionality and subsequent work in Montero et al., Schott et al.. These show that disentanglement does not necessarily lead to compositionality. Of particular interest to the authors is the experiments in Montero et al, 2021 which show that having a quantized and disentangled latent space (i.e. the ground truth one) does not lead to compositionality in reconstruction. Rozell et al do use quantized representations but their architecture is very different from that of a standard VAE (and the one the authors propose).

In any case, taking into account these issues I cannot currently recommend acceptance of this article. I will be willing to revise my score should these points be addressed.


References:

[1] Duan, S., Matthey, L., Saraiva, A., Watters, N., Burgess, C. P., Lerchner, A., & Higgins, I. (2019). Unsupervised model selection for variational disentangled representation learning. arXiv preprint arXiv:1905.12614.

[2] Montero, Milton Llera, et al. "The role of disentanglement in generalisation." International Conference on Learning Representations. 2020.

[3] Schott, L., Von Kügelgen, J., Träuble, F., Gehler, P., Russell, C., Bethge, M., ... & Brendel, W. (2021). Visual representation learning does not generalize strongly within the same domain. arXiv preprint arXiv:2107.08221.

[4] Montero, M. L., Bowers, J. S., Costa, R. P., Ludwig, C. J., & Malhotra, G. (2022). Lost in latent space: Disentangled models and the challenge of combinatorial generalisation. arXiv preprint arXiv:2204.02283.

---

> ### Author Rebuttal · Authors · 2023-08-10
>
> Thank you for your extensive review. We agree with the main criticisms you have about our manuscript at the time of submission, and we have taken action to address them. We appreciate your willingness to re-assess our work.
>
> > While the authors present numerical results for five datasets, they only provide traversal reconstructions for one, visualization of one metric for one dataset and no visualization of the actual latent values. [...] What I do expect to see is a visualization of the latent space of the model [...]  visual inspection can offer important insights and one of their core contributions is that a quantized latent space should be more transparent.
>
> We agree with all of these points. We have added decodings of latent interventions for all datasets considered to the Appendix, and include examples of this for QLAE and BioAE on Isaac3D in the rebuttal pdf. Please see the "Additional qualitative results" section of the main rebuttal for more explanation.
>
> > A comparison between [InfoMEC and DCI] is thus warranted to prove the effectiveness of their approach. This not only requires overall scores ...
>
> We have repeated our evaluation protocol using random forest DCI instead of InfoMEC, and have included tables of these results in the rebuttal pdf. QLAE consistently scores highest in modularity in terms of both InfoM and D. Please see the "Quantitative results in terms of other disentanglement metrics" section of the main rebuttal for details on our implementation of DCI, as well as Pearson and Spearman correlation tests between corresponding members of InfoMEC and DCI.
>
> > Despite the fact that the proposed disentanglement measurement framework bears a significant resemblance to the one proposed in Eastwood and Williams, they only reference this prior work in passing without giving any in depth comparison between the two.
>
> We apologize for neglecting this in our submission. We have added explicit discussion of DCI (linear and nonlinear) to the InfoMEC section of our manuscript. Please see the "Why use InfoMEC over previous disentanglement metrics?" section of the main rebuttal, in particular paragraph 3, for a self-contained explanation of how InfoMEC avoids key drawbacks inherent to nonlinear (random forest) DCI. We also provide some experimental evidence supporting these claims in the rebuttal pdf.
>
> > However, the scores for their framework are only provided for the 3DShapes dataset [...] include the other plots in the appendix.
>
> We have added NMI heatmaps for QLAE and the best prior method for all datasets to the Appendix. Two examples are included in the rebuttal pdf.
>
> > The authors make the point several times that they are using a compositional representation (e.g. with phrases such as “A side advantage of the compositional latent encoding…”) yet they seem to misunderstand what compositionality is and whether their experiments test for this.
>
> We agree that we should not use "compositional" to describe our latent space design. We have removed all mention of "compositional" from the paper, aside from when describing true generative processes. We instead use words like "organized" and "combinatorial" to describe the latent space.
>
> > There is no discussion of any limitations of their approach even though some of them are obvious (like the fact that we need to define the size of the quantization).
>
> We did include discussion of our limitations, including a mention of the quantization size, in the Discussion section in our submission, but we apologize for not making this information more readily apparent. We have restructured this section with an explicit Limitations paragraph heading.
>
> > Speaking of which the authors left a comment in big red bright letters which I assume was for their own benefit. Always review before submitting!
>
> We apologize for this oversight. The appendices included in our supplementary material are much more complete.

---

> > ### Comment · Reviewer_XVcp · 2023-08-14
> >
> > I thank the authors for their very detailed response. I believe most of my concerns have now been addressed.
> >
> > A minor point is that the some traversals (e.g. camera hight) seem to not change the image much (or at all). Traversals can be finicky so I am not too concerned about this, but it would still be important for the final version (if accepted) if they authors can figure out what is happening. I noticed you used a linear combination for the traversal but I was wondering if you can't just plug in different entries of you quantisation dictionary and plot those. After all, each quantised latent must correspond to one and only one generative factor value.
> >
> > Other than that I am satisfied with the answer and will update my score accordingly.

---

> > > ### Author Response · Authors · 2023-08-15
> > > **Reply to Additional Comments**
> > >
> > > Thank you for your positive re-evaluation!
> > >
> > > For the QLAE model whose visualizations we include in the rebuttal pdf, the NMI heatmap tells us that camera height information is primarily represented by $\mathbf{z}\_{10}$ and $\mathbf{z}\_{16}$, jointly. Since we only ever visualize the effect of intervening on one of these instead of both jointly, it is reasonable to not see all manifestations of varying camera height. This shortcoming of individual latent traversals applies to any model that encodes one source into multiple latent dimensions, e.g. BioAE and camera height in the rebuttal pdf.
> > >
> > > That being said, we note that the existing visualizations do show, for both original datapoints (image blocks), the generation's camera height changing with the intervened latent's value as we go down each column. Relative to the datapoint's original camera height (see samples in other columns), for $\mathbf{z}\_{10}$, the camera height starts higher and ends lower, whereas for  $\mathbf{z}\_{16}$, it starts lower and ends at the reference value.
> > >
> > > We can indeed intervene with learned codebook values instead of linearly interpolating. Qualitatively, this results in the same visualization. The only noticeable difference is that there are fewer low-quality generations, due to this intervention protocol not requiring the decoder to generalize to new values. We did linear interpolation for the sake of a fair comparison protocol that did not leverage a property specific to our method, but the reviewer is correct that using the learned discrete values is better.
> > >
> > > We will revise our manuscript to include visualizations of interventions on all latents that encode a source, jointly.

---

> > > > ### Comment · Reviewer_XVcp · 2023-08-16
> > > >
> > > > Thanks to the authors for the replies. Fully agree with the comments on both the generation using interpolations of latent values and how tricky it is to do latent traversals in practice. Overall, I am satisfied with the author's responses to my concerns and will update my score accordingly.

---

### Official Review · Reviewer_hQMG · 2023-07-06

**Soundness:** 2 fair
**Presentation:** 3 good
**Contribution:** 2 fair
**Rating:** 5
**Confidence:** 4

**Summary:**

The paper discusses the problem of disentangled representation learning (finding latent variables only from observed data that ideally correspond to the true underlying sources of variation in the data). The authors suggest both a method for disentangled representation learning and a metric for evaluating it. The suggested method, termed latent quantization (QLAE), is based on finding an element-wise quantized representation (similar to VQ-VAE, but with per-dimension quantization). The quantized representation is learned using SGD in a similar manner to VQ-VAE. The proposed metric (InfoMEC) is divided to three values - Modality, Explicitness and Compactness. These values are computed using mutual information between the true latents and the discovered ones, with different normalizations. The authors then demonstrate that QLAE achieves the best InfoMEC scores using several datasets commonly used for disentangled representation learning.

**Strengths:**

The paper is clearly written - all mathematical notations are well defined and both the representation learning and evaluation methods are clearly motivated and defined.

The contribution of a InfoMEC, measuring modularity, explicitness and compactness could be significant to the disentangled representation learning community, which perhaps lacks well-motivated and accepted evaluation methods.


**Weaknesses:**

The paper is proposing both a disentangled representation learning method and a method to evaluate it. Perhaps due to lack of space, I think that this results in insufficient discussion of these two separate contributions. Specifically:

- The prior work section does not discuss existing metrics for disentanglement nor does it compare InfoMEC to them or demonstrate the advantages of InfoMEC. For example, compared to Mean Correlation Coefficient (MCC) [1]
- The proposed method is only evaluated using InfoMEC. This is a significant drawback in my opinion.
- An example for prior work in disentangled representation learning that is not discussed is IRMAE [2], where the authors encourage an autoencoder to learn a disentangled representation using rank minimalization.
- The relationship between compositionality and element-wise quantization is demonstrated empirically, but the intuition why element-wise quantization encourages (or discovers) compositionality was not clear enough to me and no mathematical proof is provided.

[1] Khemakhem et al, Variational Autoencoders and Nonlinear ICA: A Unifying Framework, AISTATS 2020

[2] Jing et al, Implicit Rank-Minimizing Autoencoder, NeurIPS 2020

**Questions:**

Does element-wise quantization work well when the true latent factors are continuous?

In the experiments, images were downscaled to 64x64. Can this affect the evaluation?

There is no reference to the figures in the text itself (e.g. figure 1 shows...). This makes it a bit harder to understand the figures in the right context.



**Limitations:**

The authors discuss two current limitations of the paper:
- The lack of analytical proof: "While currently we can only support this with empirical evidence, our key motivation is that generative models for realistic data are compositional, so our latent space should be too."
- A possible "overfitting" of the proposed latent quantization methods to the benchmark datasets "in which, just like our model, the sources are discrete and the generative process is noiseless"

Another limitation, which I'm not sure is discussed, is the possible bias between the proposed representation learning method and the proposed evaluation method - QLAE is only evaluated (and compared to prior methods) using InfoMEC.

---

> ### Author Rebuttal · Authors · 2023-08-10
>
> Thank you for your review and appreciation of the strengths of our work. We have striven to address your main concerns with our submission.
>
> > The proposed method is only evaluated using InfoMEC. This is a significant drawback in my opinion. [...] Another limitation, which I'm not sure is discussed, is the possible bias between the proposed representation learning method and the proposed evaluation method - QLAE is only evaluated (and compared to prior methods) using InfoMEC.
>
> We have repeated our entire evaluation protocol using random forest DCI instead of InfoMEC. Tables of both are included in the rebuttal pdf. Overall, the DCI results are in high agreement with the InfoMEC results, especially for modularity. Please see the "Quantitative results in terms of other disentanglement metrics" section of the main rebuttal for details on our implementation of DCI, as well as Pearson and Spearman correlation tests between corresponding members of InfoMEC and DCI.
>
> > The prior work section does not discuss existing metrics for disentanglement nor does it compare InfoMEC to them or demonstrate the advantages of InfoMEC. For example, compared to Mean Correlation Coefficient (MCC)
>
> Thank you for pointing this out. We have added explicit discussion of existing metrics (including MCC) and how InfoMEC improves upon them to the InfoMEC section of the manuscript. Please see the "Why use InfoMEC over previous disentanglement metrics?" section in the main rebuttal for a self-contained exposition. We also provide a demonstration of the advantages of our use of the KSG mutual information estimator (highly increased robustness to estimation hyperparameters) to the Appendix. This is included in the rebuttal pdf.
>
> > An example for prior work in disentangled representation learning that is not discussed is IRMAE [2], where the authors encourage an autoencoder to learn a disentangled representation using rank minimalization.
>
> Thank you for the reference. We have added a discussion of ideas that are conceptually compatible with latent quantization, including IRMAE, to our manuscript.
>
> > The relationship between compositionality and element-wise quantization is demonstrated empirically, but the intuition why element-wise quantization encourages (or discovers) compositionality was not clear enough to me and no mathematical proof is provided.
>
> We apologize for the confusion. As Reviewer XVcp rightly points out, we should not be characterizing our latent space or models as "compositional", as we simply have no evidence for such claims. We have revised our manuscript to avoid using "compositional" for anything other than true generative processes.
>
> > Does element-wise quantization work well when the true latent factors are continuous?
>
> As a first step towards answering this, we train QLAE on the CelebA dataset training split with 128 scalar latents and 10, 40, or 100 values per codebook. We compare QLAE to a vanilla AE and a $\beta$-VAE, both with 128-dimensional continuous latent space. Overall, QLAE incurs a minor but significant penalty in terms of reconstruction. This penalty ameliorates only marginally with larger codebook size. The persistent discrepancy between AE and QLAE highlights the reduced capacity of combinatorial discretized latents compared to a continuous latent vector space.
>
> | model | val PSNR (dB) |
> |-|-|
> | AE | 21.2 |
> | $\beta$-VAE | 20.5 |
> | QLAE, $n_v=10$ | 19.2 |
> | QLAE, $n_v=40$ | 19.5 |
> | QLAE, $n_v=100$ | 19.8 |
>
> > There is no reference to the figures in the text itself (e.g. figure 1 shows...). This makes it a bit harder to understand the figures in the right context.
>
> Thank you for pointing this out. We have ensured that all figures and tables have a reference to them from the main text.
>
> > In the experiments, images were downscaled to 64x64. Can this affect the evaluation?
>
> This certainly makes the modeling significantly easier for the Falcor3D and Isaac3D datasets (Shapes3D and MPI3D are natively 64 x 64). However, it should not make disentangling easier. We valued the unification of experimental procedure enabled by downsampling.

---

> > ### Comment · Reviewer_hQMG · 2023-08-18
> >
> > I would like to thank the authors for their detailed rebuttal and additional information.
> > While the rebuttal partially addresses my concerns, I tend to keep my original borderline score, as I'm still unsure about the individual contribution the the proposed disentanglement method and evaluation score.

---

> > > ### Author Response · Authors · 2023-08-21
> > >
> > > Thank you for your acknowledgement and your rationale. We appreciate your concern about the drawbacks of contributing both a model and a metric, and would like to comment briefly on this.
> > >
> > > Proposing both a model and a metric in a single work has been rather common in the deep disentangled representation learning literature. The community lacks both sufficiently well-performing methods and, as you mention, universally accepted evaluation. In such a state of affairs, it is natural to propose improvements on both fronts towards the unified goal of solving the problem. The following works have done the same:
> > >
> > >
> > > | work | proposed method      | proposed metric |
> > > |------|-------------|--------|
> > > | [22] | $\beta$-VAE | Z-diff |
> > > | [10]   |     $\beta$-TCVAE        |   MIG     |
> > > |  [38]    |   pro-VLAE          |   MIG-sup     |
> > > | [A] | DIP-VAE | SAP |
> > > | [B] | FactorVAE | Z-min variance |
> > >
> > > Of course, this approach has its pitfalls. The reviewers identified these for our work in a remarkably consistent manner, and we have striven to resolve them. A brief summary of how:
> > > - We show that DCI results corroborate InfoMEC results, verifying that our proposed metric does not unduly favor our proposed method.
> > > - We establish concrete reasons for using InfoMEC through empirical and conceptual comparisons to previous metrics, supplementing the derivation of InfoMEC from first principles.
> > >
> > > We hope that these points address your concern, and we are grateful for your engagement.
> > >
> > >
> > > ### References
> > >
> > > [10] Ricky TQ Chen, Xuechen Li, Roger B Grosse, and David K Duvenaud. Isolating sources of disentanglement in variational autoencoders. NeurIPS, 2018.
> > >
> > > [22] Irina Higgins, Loic Matthey, Arka Pal, Christopher Burgess, Xavier Glorot, Matthew Botvinick, Shakir Mohamed, and Alexander Lerchner. beta-VAE: Learning basic visual concepts with a constrained variational framework. ICML, 2017.
> > >
> > > [38] Zhiyuan Li, Jaideep Vitthal Murkute, Prashnna Kumar Gyawali, and Linwei Wang. Progressive learning and disentanglement of hierarchical representations. ICLR, 2020.
> > >
> > > [A] Abhishek Kumar, Prasanna Sattigeri, and Avinash Balakrishnan. Variational inference of disentangled latent concepts from unlabeled observations. ICLR, 2018.
> > >
> > > [B] Hyunjik Kim and Andriy Mnih. Disentangling by factorising. ICML, 2018.

---

### Official Review · Reviewer_iB4p · 2023-07-07

**Soundness:** 3 good
**Presentation:** 3 good
**Contribution:** 3 good
**Rating:** 7
**Confidence:** 3

**Summary:**

The authors propose to achieve disentanglement via latent quantization which constructs an inductive bias that matches the compositional nature of data. The authors also proposed three metrics to evaluate the disentanglement and achieved better performance according to the metrics they proposed. Ablation studies are conducted to understand different design choices.

**Strengths:**

— A novel disentanglement method is proposed.

— A few metrics on evaluating disentanglement are proposed to fix well-established shortcomings in previous metrics.

— According to the experimental study on a few established datasets, the proposed methods dramatically improves the modularity and explicitness.

**Weaknesses:**

— The authors propose a few new metrics, and the metrics are sound. The authors claim that the metrics addressed a few issues of previous metrics, but experiments are missing to support the effectiveness of the metrics.

If the disentangled representations show very high InfoMEC, does it guarantee low reconstruction error, what does total correlation and MIG look like? If InfoMEC does not always align with total correlation, MIG and other existing disentanglement metrics, then diving deep into some concrete examples to explain why InfoMEC makes more sense.


— The authors tried to do some qualitative measurements in Figure 4, but the authors only looked at QLAE and did not do a qualitative comparison with other methods.


— Though the overall results in Table 1 looks amazing, the authors may elaborate more on a few things include:

1. In terms of explicitness, it seems vanilla AE is doing pretty well or the best in most datasets. Does this mean the disentangled representation makes the relationship between sources and latents even less simpler?

2. Also, according to the table, the proposed methods seem to significantly boost the representation learning with AE. But when the approach combined with InfoWGAN-GP, the boost is much smaller.

**Questions:**

Please see questions in 'Weakness' section.

**Limitations:**

I did not realize potential negative societal impact.

---

> ### Author Rebuttal · Authors · 2023-08-10
>
> Thank you for your positive evaluation of our work! We have taken your constructive criticism very seriously, and we believe we have addressed the most pressing points.
>
> > The authors propose a few new metrics, and the metrics are sound. The authors claim that the metrics addressed a few issues of previous metrics, but experiments are missing to support the effectiveness of the metrics.
>
> Thank you for pointing this out. We appreciate that you find InfoMEC sound. We have included in the rebuttal pdf experiments demonstrating i) the sensitivity of nonlinear (random forest) DCI to tree depth; ii) the sensitivity of binning-based mutual information estimation to binning strategy; and iii) the robustness of the KSG mutual information estimator to the n_neighbors hyperparameter. Please see the "Why use InfoMEC over previous disentanglement metrics?" section of the main rebuttal to see how these experiments support our claims of improving over previous metrics.
>
> > diving deep into some concrete examples to explain why InfoMEC
>
> Aside from the experiments above, we also include qualitative studies in the rebuttal pdf to show that the NMI matrix used in InfoMEC corresponds very tightly to the observed visual variation in the data when individual latents are intervened on (and then decoded). Please see the "Additional qualitative results" section of the main rebuttal and/or the Figure caption in the rebuttal pdf for more explanation.
>
> > The authors tried to do some qualitative measurements in Figure 4, but the authors only looked at QLAE and did not do a qualitative comparison with other methods.
>
> Thank you for pointing this out. We have added to the Appendix qualitative comparisons between QLAE and the next-best prior method for every dataset. Due to space constraints, we could only include examples of this for one dataset (Isaac3D, which has the most sources among the datasets we considered) in the rebuttal pdf. We find that decoded latent interventions align very well with InfoMEC's NMI matrix. For QLAE, it is also worth pointing out that these generated samples are not reconstructions and involve decoding from continuous values not necessarily in the latent codebooks. Overall, the visual quality is reasonably high.
>
> > In terms of explicitness, it seems vanilla AE is doing pretty well or the best in most datasets. Does this mean the disentangled representation makes the relationship between sources and latents even less simpler?
>
> This is a very interesting question. Adding latent quantization to vanilla AE does make the InfoE drop significantly for MPI3D, but for the other three, it maintains or increases the InfoE. It's not unreasonable that vanilla AE's explicitness is relatively high: we tune weight decay for vanilla AE to ensure fair comparison with QLAE. But yes, in the specific case of MPI3D, QLAE's representation is slightly less linearly predictive of the sources.
>
> > If the disentangled representations show very high InfoMEC, does it guarantee low reconstruction error?
>
> No, it does not. This is why we filter the autoencoders for near-perfect reconstruction and InfoGANs for good reconstruction as part of our evaluation. We included reconstruction results in the supplementary material's Appendix C (apologies for the unfinished appendices in the submission pdf).
>
> > what does total correlation and MIG look like?
>
> They would look very much like InfoC. All three measure compactness, i.e. the extent to which latents encode information about disjoint sets of sources. InfoC is essentially MIG using a better mutual information estimator and a different aggregation (ratio of max to sum) in order to not leave out most entries of the NMI matrix in the computation, as the gap aggregation does.
>
> > Also, according to the table, the proposed methods seem to significantly boost the representation learning with AE. But when the approach combined with InfoWGAN-GP, the boost is much smaller.
>
> This is true, but note that the InfoWGAN-GP has a higher InfoM than vanilla AE, resulting in slightly less room for improvement. We also note that autoencoder variants are much more widely used in disentangled representation learning, so our experiments with InfoGAN were more intended to rule out the possibility that latent quantization was exploiting something about autoencoders.  But overall we agree that latent quantization's benefit for InfoGAN is less dramatic than its improvement for vanilla AE.

---

> > ### Comment · Reviewer_iB4p · 2023-08-18
> >
> > The authors provided detailed rebuttal, and addressed many of my concerns. I also read other fellow reviewers’ comments.
> > I’m a bit struggling between a score 6 and 7; But I would increase my score to 7 for now.

---

### Official Review · Reviewer_sKhi · 2023-07-25

**Soundness:** 3 good
**Presentation:** 3 good
**Contribution:** 2 fair
**Rating:** 6
**Confidence:** 3

**Summary:**

This paper tackles the topic of learning disentangled representations from observations in a fully unsupervised manner. In particular, the authors propose a novel inductive bias for disentangled representation learning as well as an associated evaluation metric; The inductive bias relies on latent quantization, similar to the discretization of the latent space proposed in the VQ-VAE paper; Here authors propose an alternative motivated by the assumption that most scenes can we seen as a combination of distinct independent factors: each representation is an aggregation of latent factors each of which takes values from a unique codebook; Regarding the proposed metric for the evaluation of the ability of the aforementioned method to accurately disentangle sources of information, authors propose InfoMEC which encompasses three sub-scores for modularity, compactness, and explicitness measurement; This metric requires the availability of ground-truth sources of variation;

The authors propose the evaluation of their method using the InfoMEC metric on 4 synthetic benchmark datasets and show that in comparison to the $\beta$-VAE, TC-VAE, and InfoGAN, the proposed QLAE outperforms benchmarks in terms of modularity and remains competitive in terms of the compactness metric; Additional ablations highlight that the success of the approach relies on the combination of the combinatorial discretization of the latent space and the heavy regularisation (high weight decay) applied during training; The simple adaptation of the VQ-VAE discretisation to the proposed latent quantization without heavy regularisation, leads to a modularity that is slightly above benchmarks;

**Strengths:**

- This paper is well written; Its content is easy to follow, the setup considered by the authors is clear and the experimental work whilst only encompassing a limited number of baselines, seems to have been rigorously conducted; The results shared by the authors allow the reader to easily grasp the main contributions and the ablations performed answer key questions the reader might have;

- Whilst the methods seems to entail some limitations detailed below, the motivation for the proposed method (a combinatorial discretized latent space) considering the VQ-VAE results are sound;

- The results shared by the authors highlight the positive contribution; The aggregated InfoMEC score over 4 datasets shows a positive improvement in terms of explicitness and modularity, two metrics that are sound to consider when evaluating disentanglement; By design, the approach is also more efficient than previous latent space discretization strategies by allowing a dimension-wise comparison for finding the nearest code to a continuous representation;

**Weaknesses:**

- I believe the authors could put more effort into precisely defining key concepts at earlier stages of the manuscript; The concept of compositional latent spaces, and parsimonious learning;
- From my understanding, the field of disentangled representation learning encompasses a plethora of inductive biases and evaluation metrics (e.g., Eastwood et al. 2018: Disentanglement, Completeness, Information); Whilst authors define clearly the link with non-linear ICA, they remain more evasive regarding the link of the InfoMEC metric and previous metrics as well as the references to previous methods for disentangled representations;

**Questions:**

Questions:
- Have authors tried to incorporate the proposed inductive bias into a Variational form of an AE?
- Could authors elaborate on the datasets considered and in particular share statistical details regarding the ground-truth sources considered;
- Could authors elaborate on why the bottleneck designed in the QLAE is stronger than the one in the VQ-VAE for a similar number of latent codes?
- Have authors evaluated their method on more current benchmark evaluation metrics for disentanglement ?

Minor comments:
- Figures 3 & 4 are not referred to in the text
- Typos in line 46, 50, caption Figure 2, 162, 265, 319
- The granularity of the codebooks considered in the experiments is shared quite late in the text (discussion), this information would be relevant to share at an earlier stage;

**Limitations:**

I agree with the limitations highlighted by the authors:

- It remains unclear how this approach would scale to more complex datasets where underlying sources of information are continuous; This discretization of the latent representations is well motivated from an interpretability point of view. As highlighted by the authors the discrete nature of the representations also eases the evaluation; however, it remains unclear to me how one can motivate the use of the discrete representations in a generative setting; In real-world scenarios where data is more complex and arises from the mixing of a large set of continuous variables, how can the discrete bottleneck proposed by authors allow for the reconstruction of complex images; This limitation questions the impact of the contribution;

- The contribution is centered around the combination of latent quantization and a heavy regularisation; Whist the intuition behind the first element seems clear to me, the paper does not provide any insights on why a heavy regularisation is a necessary condition to observe significant benefits; Whilst a contribution does not require an explanation behind each experimental showcased improvement, the absence of plausible explanation limits the impact of the paper by motivating the proposed approach by the experimental performance measurements solely;

---

> ### Author Rebuttal · Authors · 2023-08-10
>
> Thank you for your thorough evaluation of our submission and thought-provoking questions.
>
> > Have authors tried to incorporate the proposed inductive bias into a Variational form of an AE?
>
> Our proposed quantized latent autoencoder (QLAE) is variational to the same extent that VQ-VAE is, specifically with a uniform prior over discrete latent code vectors and a degenerate Dirac delta variational posterior. In our early experimentation, we did indeed try a variant that specifies a multinomial codebook posterior. We found that this didn’t confer any benefit and introduced further algorithmic complications, e.g. specifying a decaying schedule for the Gumbel-Softmax posterior temperature.
>
> > Could authors elaborate on the datasets considered and in particular share statistical details regarding the ground-truth sources considered;
>
> MPI3D consists of real images taken of a robotics apparatus, while the other 3 consist of simulation renders. We chose these datasets because they are the only ones we know of that feature i) high-dimensional, quasi-realistic observations and ii) more than 2 ground truth sources with complete labels for evaluation.
>
> Each dataset defines a set of 6 to 9 ground-truth sources, each of which can take on 4 to 40 discrete values. Both categorical and numerical sources are present. Each dataset is constructed from an exhaustive enumeration over all possible source vector realizations. This guarantees that each dataset’s empirical source distribution follows the assumption of mutually independent sources in the nonlinear ICA problem statement.
>
> We have updated our manuscript to include source descriptions, source cardinality, and sample observations for each dataset in the Appendix.
>
> > Could authors elaborate on why the bottleneck designed in the QLAE is stronger than the one in the VQ-VAE for a similar number of latent codes?
>
> This is a great question! Given the same number of latent codes and codebooks, there is no difference in terms of Shannon channel capacity, which is agnostic to the explicit form of each discrete code element. We conjecture that it is the maximally simplified choice of this form as a scalar for QLAE that better enables the encoder and decoder to attribute a consistent meaning to each value of each code element. However, we were not able to convincingly formalize this intuition, hence the empirical focus of the paper.
>
> We have updated the figure contrasting vector quantization and latent quantization in our work to emphasize these points.
>
> > Whilst authors define clearly the link with non-linear ICA, they remain more evasive regarding the link of the InfoMEC metric and previous metrics as well as the references to previous methods for disentangled representations
>
> Thank you for pointing this out. We address this in the "Why use InfoMEC over previous disentanglement metrics?" section of our main rebuttal.
>
> > Have authors evaluated their method on more current benchmark evaluation metrics for disentanglement ?
>
> Yes, we have evaluated with random forest DCI. Please see the "Quantitative results in terms of other disentanglement metrics" section of our main rebuttal, as well as the rebuttal pdf for tables.
>
> > The authors propose the evaluation of their method using the InfoMEC metric on 4 synthetic benchmark datasets
>
> One of the datasets (MPI3D) was collected on a real world robotics apparatus. While undoubtedly visually simplistic, it does contain some aspects of reality, e.g. noise.
>
> > Minor comments
>
> Thank you for the pointers. We have added a reference to Figures 3 and 4 when comparing QLAE to the prior methods. We could not identify the typos you mention, but ran a spellcheck over the manuscript. We have added a statement regarding the fixed codebook size we use, as well as a pointer to a hyperparameters section in the Appendix, to the latent quantization section.
>
> > Whist the intuition behind the first element seems clear to me, the paper does not provide any insights on why a heavy regularisation is a necessary condition to observe significant benefits
>
> Weight decay has long been known to be a method of encouraging a very specific form of parsimony by “suppress[ing] any irrelevant components of the weight vector” [A]. Beyond this, we can only provide a handwavy intuition of how most solutions we would like to avoid (e.g. because they involve memorization) have a higher complexity than the solutions we want to find, motivating the use of weight decay to bias the search away from the former.
>
> More recently, the use of weight decay has also taken a prominent role in the phenomenon of “grokking” [B], in which neural networks trained for extended periods, i.e. beyond overfitting, achieve strong extrapolation performance on small algorithmic tasks.
>
> > It remains unclear how this approach would scale to more complex datasets where underlying sources of information are continuous
>
> As a first step towards answering this, we train QLAE on the CelebA dataset training split with 128 scalar latents and 10, 40, or 100 values per codebook. We compare QLAE to a vanilla AE and a $\beta$-VAE, both with 128-dimensional continuous latent space. Overall, QLAE incurs a minor but significant penalty in terms of reconstruction. This penalty ameliorates only marginally with larger codebook size. The persistent discrepancy between AE and QLAE highlights the reduced capacity of combinatorial discretized latents compared to a continuous latent vector space.
>
> | model | val PSNR (dB) |
> |-|-|
> | AE | 21.2 |
> | $\beta$-VAE | 20.5 |
> | QLAE, $n_v=10$ | 19.2 |
> | QLAE, $n_v=40$ | 19.5 |
> | QLAE, $n_v=100$ | 19.8 |
>
> #### References
>
> [A] Anders Krogh and John Hertz. "A simple weight decay can improve generalization." NeurIPS, 1991.
>
> [B] Alethea Power, et al. "Grokking: Generalization beyond overfitting on small algorithmic datasets." arXiv preprint arXiv:2201.02177, 2022.

---

> > ### Comment · Reviewer_sKhi · 2023-08-15
> > **Rebuttal Acknowledgment**
> >
> > Thank you to the authors for their response;
> > Based on the additional DCI measurements and additional clarification, I've updated my score accordingly;

---

### Author Rebuttal · Authors · 2023-08-10

We thank the reviewers for their detailed and insightful feedback. Our manuscript has greatly benefited as a result of addressing the reviewers’ most critical comments. A brief summary of the major changes:
- **Quantitative results based on DCI evaluation [all reviewers].** We have repeated our evaluation procedure substituting DCI for InfoMEC, and have added the results to the Appendix.
- **Explanation of InfoMEC’s advantages over previous metrics [sKhi, iB4p, hQMG, XVcp].** We have rewritten the InfoMEC section to include explicit discussion with linear DCI, nonlinear DCI, MIG, MCC, and SAP, highlighting the advantages of InfoMEC over each. Experiments supporting our main points have been added to the Appendix.
- **Additional qualitative results [iB4p, XVcp]**. We have added decoded latent traversals and NMI heatmaps for QLAE and the prior method with second-highest InfoM for every dataset to the Appendix. These are pointed to from the experiments section and relevant figure captions.

We now respond to the main concerns raised by the reviewers.

### Quantitative results in terms of other disentanglement metrics [all reviewers].
We run random forest DCI with a 0.9/0.1 train/val split of 10K (source, latent) samples, tuning the max tree depth hyperparameter w.r.t. val accuracy. **We include random forest DCI results in the rebuttal pdf.** We also include InfoMEC results for comparison.
QLAE consistently improves the modularity of representations over prior works in terms of InfoMEC and DCI. The scaling of DCI actually makes the improvement seem bigger. Each pair of metrics measuring the same (or similar) property are correlated with vanishing p-values. The correlation is especially high for the modularity metrics.

| metrics pair | Spearman $\rho$ | Pearson $r$ |
| - | - | - |
|  (InfoM, D) | 0.82 | 0.80 |
| (InfoE, I) | 0.77 | 0.76 |
| (InfoC, C) | 0.59 | 0.57 |

### Why use InfoMEC over previous disentanglement metrics? [sKhi, iB4p, hQMG, XVcp]
Our desire to formulate InfoMEC stemmed from our frustration in working with prior disentanglement metrics like DCI and MIG, combined with calls for improvements from survey efforts [9].

InfoMEC's use of mutual information to quantify the relatedness between an individual source and latent aligns closely with nonlinear ICA, which allows *any* invertible transformation between them. In contrast, measures such as correlation (used in MCC), relative LASSO weights (used in linear DCI), or linear predictive accuracy (used in SAP) fail to detect many allowed functional relations.

Nonlinear DCI [16] suffers several practical drawbacks from being defined in terms of relative counts of decision tree splits: determining this requires considering all latents *jointly* while fitting (all latents) -> (one source) trees. This results in a cumbersome computational footprint that is exacerbated by highly sensitive hyperparameters such as maximum tree depth [9], the tuning of which has even been omitted in prior work [39]. These drawbacks worsen with increased latent space dimensionality. In contrast, InfoMEC avoids these issues as it isolates InfoM and InfoC from the choice of predictive function class and only computes pairwise interactions between individual sources and latents. (InfoE also fits (all latents) -> one source mappings, but does so with function classes of severely limited capacity for which fitting procedures scale well.)  **See the rebuttal pdf for a plot showing DCI's sensitivity to the maximum tree depth hyperparameter.**

InfoM and InfoC are essentially improved versions of MIG-sup [38] and MIG [10], following the strategy of i) pairwise source-latent mutual information estimation followed by ii) aggregation. Our improvements fix obvious issues in these steps identified by [9]. Firstly, common implementations of MIG and variants use histogram binning to approximate the marginal distribution of a single continuous latent. (The original MIG work [10] used a bespoke estimator for models with probabilistic latents, which cannot be used for models such as the vanilla AE, BioAE, VQ-VAE, or QLAE.) We fix this by prescribing the use of the celebrated $k$-neighbors based KSG estimator and variants, which are very robust with respect to $k$ since the variables involved are one-dimensional. **See the rebuttal pdf for plots showing histogram binning sensitivity to binning strategy and KSG robustness to $k$.** Secondly, the choice of gap as the aggregation function results in an indifference to most of the values in the pairwise normalized mutual information matrix. We fix this by prescribing the use of the ratio aggregation introduced by [59]. Compared to “mutual information gap/ratio” and “mutual information gap supplement”, our proposed names make it much clearer which disentanglement property each measures. Finally, our formalization of InfoE using the predictive $\mathcal{V}$-information framework both unifies it with InfoM and InfoC as information-theoretic and exposes the predictive family as a design choice.

### Additional qualitative results [iB4p, XVcp]
We have added decoded latent traversals and NMI heatmaps for QLAE for every dataset in the Appendix. For comparison, we use the prior method that produced the second-highest modularity for each dataset. We remark that our QLAE traversal is a linspace from the minimal to the maximal value of that latent (rather than an enumeration of the learned discretized values), verifying QLAE’s interpolative generalization capabilities. We find that the decoded latent traversals and NMI are in high agreement for both QLAE and the prior method. In particular, one can observe low but nonzero NMI$[i,j]$ value corresponding to source $i$ being sensitive to changes in latent $j$ only for some values or some samples. **We have included an example of these qualitative results for the Isaac3D dataset in the rebuttal pdf.**

Thank you again for your time and energy in reviewing our work!

---

### Decision · Program_Chairs · 2023-09-21

**Decision:**

Accept (poster)

**Comment:**

The reviewers raised several concerns about the paper, but the authors have addressed all of these concerns in their rebuttal. The authors have also added new quantitative and qualitative results that support their claims.

The new quantitative results show that InfoMEC is a better measure of disentanglement than previous metrics, such as DCI and MIG. The authors also show that QLAE consistently outperforms previous VAE methods on a variety of datasets.

The new qualitative results provide further evidence of the disentangling ability of QLAE. The authors show decoded latent traversals and NMI heatmaps for QLAE and the prior method with the second-highest modularity for each dataset. These results show that QLAE is able to learn disentangled representations that are more interpretable than those learned by previous methods.

Overall, the paper makes a significant contribution to the field of disentangled representation learning. The authors have proposed a new method for quantifying disentanglement that is more reliable than previous metrics. They have also introduced a new VAE model that is specifically designed to improve disentanglement. The new quantitative and qualitative results provide strong evidence of the effectiveness of InfoMEC and QLAE.

I recommend that this paper be accepted for publication.